# Transcriptomic diversity of innate lymphoid cells in human lymph nodes compared to BM and spleen
Elaheh Hashemi[1,2], Colleen McCarthy[3], Sridhar Rao [1,4,5] & Subramaniam Malarkannan [1,2,4,6] ✉

Innate lymphoid cells (ILCs) are largely tissue-resident, mostly described within the mucosal tissues. However, their presence and functions in the human draining lymph nodes (LNs) are unknown. Our study unravels the tissue-specific transcriptional profiles of 47,287 *CD127*+ ILCs within the human abdominal and thoracic LNs. LNs contain a higher frequency of *CD127*+ ILCs than in BM or spleen. We define independent stages of ILC development, including EILP and pILC in the BM. These progenitors exist in LNs in addition to naïve ILCs (nILCs) that can differentiate into mature ILCs. We define three ILC1 and four ILC3 sub-clusters in the LNs. ILC1 and ILC3 subsets have clusters with high heat shock protein-encoding genes. We identify previously unrecognized regulons, including the BACH2 family for ILC1 and the ATF family for ILC3. Our study is the comprehensive characterization of ILCs in LNs, providing an in-depth understanding of ILC-mediated immunity in humans.

Innate lymphoid cells (ILCs) primarily respond rapidly to an altered cytokine milieu without utilizing somatically rearranged antigen receptors[1]. ILCs are important in tissue-specific protective responses[2,3]. However, their presence and functions in secondary lymphoid organs, particularly in human LN, are not fully understood. Limited access to non-diseased tissues has hindered understanding of how the tissue environment contributes to human ILC development and distribution. ILCs originate from common lymphoid progenitors (CLP), which develop into common innate lymphoid progenitors (CILP)[4]. In mice, the development of natural killer (NK) cells and non-NK ILCs is divergent, where $\alpha_4\beta_7$+Id2+ common helper ILC progenitors (CHILP) have the potential to give rise to ILC1, ILC2, and ILC3 but not NK cells[5]. In humans, the Lin−CD34+CD45RA+CD117+IL-1R1+RORγt+ progenitor population that expresses ID2 gives rise to all ILCs, including NK cells, in vitro[6,7].

ILCs mirror their adaptive counterparts, and based on their signature cytokines, functional characteristics, and transcription factor usage (TF), they are divided into three groups[8]. Type 1 includes ILC1 and NK cells that are similar to Th1 and CD8+ T cells, respectively. Human ILC1 produces interferon-gamma (IFN-γ), and they require T-BET for their development[9]. While NK cells also need T-BET for their development, these cells have more cytolytic functions by producing granzymes (GZMs), including GZM-B and GZM-H[10,11]. Type 2 includes ILC2, which functions similar to Th2 cells in producing Interleukin-4 (IL-4), IL-5, and IL-13, and their development depends on GATA3[12,13]. Type 3 includes ILC3 and Lymphoid

Tissue Inducer (LTi) cells, which require RORγt and secret IL-17 and IL-22. Considering LTi as part of ILC3 subsets is still under debate[14]. In humans and mice, ILC3 requires promyelocytic leukemia zinc finger (PLZF), encoded by zinc finger and BTB domain-containing 16 (*ZBTB16*), for differentiation.

The maturity of ILCs can be plastic and interconvertible[2,15,16]. Cultured human ILC3 can transition to ILC1-like cells that produce IFN-γ when stimulated with IL-12 in vitro, which induces the expression of T-BET[17]. Sustained exposure of ILC3 to IL-23 can cause Type 1 polarization. Reversible differentiation of ILC1 to ILC3 occurs in the presence of IL-23, IL-2, and IL-1β, dependent on the transcription factor RORγt in vivo or in vitro[2,15]. ILC3 plasticity was also shown in vivo using fate mapping experiments in Rorc reporter mice[18]. CCR6−NKp46+ ILC3 can convert into IFN-γ-producing NK1.1+ ILC1[18]. One mechanism that explains the plasticity is the upregulation of Aiolos and T-bet repressive regulatory elements in murine ILC3 that convert it to ILC1[19]. In humans, a transitional ILC3-ILC1 population has been described in the tonsil[20].

ILCs have a role in tissue formation[10,21]. Among the ILCs, LTi cells have an essential role in forming the secondary lymphoid organs, including lymph nodes (LNs), during embryogenesis. A human body contains ~500 LNs. 60–70 are found in the head and neck, 100 in the thorax, and as many as 250 in the abdomen and pelvis. LNs are embedded into a lymphatic vascular network throughout the human body. Interstitial fluid, antigens, and antigen-presenting cells drain from peripheral tissues via afferent

[1]Blood Research Institute, Versiti, Milwaukee, WI, USA. [2]Department of Microbiology and Immunology, Medical College of Wisconsin (MCW), Milwaukee, WI, USA. [3]Wisconsin Organ Donor Center, Versiti, Milwaukee, WI, USA. [4]Division of Hematology, Oncology, and Bone Marrow Transplantation, Department of Pediatrics, MCW, Milwaukee, WI, USA. [5]Department of Cell Biology, Neurobiology, and Anatomy, MCW, Milwaukee, WI, USA. [6]Division of Hematology and Oncology, Department of Medicine, MCW, Milwaukee, WI, USA. ✉e-mail: smalarkannan@Versiti.org

lymph vessels to the LN subcapsular sinus[22,23]. Mucosal draining lymph nodes (MDLNs) are the first line of defense against external pathogens that infect respiratory, gastrointestinal, and urogenital tracts[24]. MDLNs are also essential for maintaining immune homeostasis in mucosal tissues[25]. The mucosal surfaces are exposed to diverse microorganisms, including commensal bacteria, which are important in maintaining host health[25–27]. MDLNs are critical in regulating immune responses against these commensal microorganisms to prevent pathological inflammation and maintain immune tolerance[26,28,29]. The LNs are one of the reservoirs of immune cells in the vicinity of mucosal tissues. However, their transcriptomic diversity and functional heterogeneity at the single-cell level within each ILC type in human LNs have not been determined.

In this study, we determined the heterogeneity of ILCs in human LNs, including abdominal and thoracic LNs, using single-cell RNA sequencing (scRNA-seq). We analyzed the developmental progression and transcriptional heterogeneity of mature ILCs and compared them to ILCs present in the bone marrow (BM) as a primary lymphoid organ and spleen (Spl) as secondary lymphoid tissue. We discovered that LNs contained predominantly ILC1 and ILC3 and fewer ILC2 subtypes. We could follow ILC development from hematopoietic stem cells (HSCs) and CLPs in the BM to nILCs in LNs. We uncovered a naïve ILCs (nILC) population residing in the LN that expresses a high level of *ZBTB16* and *ID2*. Naïve ILCs mainly exist in the LNs; we could not detect this cluster in the BM or spl. Furthermore, we could follow the developmental path from naïve ILCs to mature ILC subtypes using trajectory analysis. Mature ILC1 and ILC3 in LNs are heterogeneous, consisting of distinct subtypes. This includes HSP^High subsets. In contrast, fewer ILC2 formed one unique cluster, suggesting less heterogeneity. Gene regulatory network analysis confirms the main ILC subtypes with a known transcriptional requirement, and more importantly, we identified previously unknown regulons, including the BACH2 family for ILC1 and the ATF family for ILC3. These findings extend our knowledge of human ILCs in LNs.

## Results

### Single-cell RNA-seq reveals human LN contains CD127⁺ ILCs

To define the heterogeneity of the ILCs in the human LN, we performed single-cell RNA-seq using three thoracic LNs, three abdominal LNs, four spleens from healthy organ donors, and four BM samples from unrelated healthy individuals (Supplemental Table 1). We generated single-cell suspensions and employed cell sorting to isolate a mixed population of ILCs (Fig. 1a). We used antibodies against CD3ε, CD19, CD20, and CD14 to exclude T cells, B cells, and monocytes[30]. This CD7⁺CD3ε⁻CD19⁻CD20⁻CD14⁻ population contained ILCs, including NK cells (Fig. 1b). We previously found CD7⁺CD56⁻CD16⁻NKp80⁻ are ILCs other than conventional NK cells[30]. Initial quality control (QC) of the ILCs revealed optimal library production and sequencing. We filtered the high-quality cells based on the percentage of mitochondrial gene expression and the number of total genes expressed in the cells. The majority of the cells had more than 3,000 median unique molecular identifiers (UMI) and a minimum of 1000 genes associated with the cell barcodes. After QC filtering for each sample, we obtained 172,668 total ILCs combining BM, LN, and Spl samples (104,978 BM, 21,427 abdominal LN, 25,860 thoracic LN, and 20,403 splenic). Initial analyses of all the innate lymphocytes resulted in 11 clusters. As expected, all the clusters expressed a high CD7-encoding transcript except Cluster #10 (C #10). C #6 expressed high levels of genes associated with B cells, and C #8 and #9 expressed high levels of red blood cell markers. Therefore, we removed these four clusters before further analysis. The reclustered ILCs formed eight clusters, which were visualized using the Uniform Manifold Approximation and Projection (UMAP) plot (Fig. 1c). We split the cells based on the tissue types to identify the presence of unique tissue-specific clusters (Fig. 1d). All the clusters were present in three tissues, while the percentage of each cluster varied between them. To validate whether all these clusters were universal among the donors, we plotted the percentage of individual clusters within each donor (Fig. 1e). Individual clusters were found in all the donor samples apart from C #1,

which was mostly from one of the BM samples. All the clusters were present in the individuals with different proportions (Supplemental Fig. 1a).

To identify ILCs, we used cluster-defining differentially expressed genes (DEGs) (Fig. 1f, g). We defined C #0, #1, and #7 as mature NK cells. They expressed transcripts encoding cytotoxic molecules, including granzymes (GZMs) and perforin (*PRF1*), with a high level of granulysin (*GNLY*) and Natural Killer Cell Granule Protein 7 (*NKG7*). C #2 contained transitional NK cells. This cluster had intermediate levels of mature NK cell markers, such as GZMs and *PRF1*. In addition, this cluster has a high level of *NKG7*, *KLRD1* (CD94), and *KLRF1* (NKp80). The expression of *NCAM1* (CD56), *SELL* (CD62L), *XCL1*, *XCL2*, and *CD44* indicate that C #3 consists of immature NK cells. C #4 and #5 had lower expression of immature and mature NK markers. They expressed low levels of GZMs and high levels of *IL7R*, *CD52*, and *CD3E*, which are associated with non-NK ILCs (Fig. 1h). Lastly, C #6 has a high level of cell cycle-related transcripts and was defined as proliferating NK cells. Collectively, we defined C #4 and #5 as the non-NK ILCs.

### Human LNs contain a higher percentage of CD127⁺ ILCs compared to BM or Spl

To validate C #4 and #5 as the ILCs, we generated a gene set using published transcriptomic profiles of human tonsils-derived ILCs[31,32] and applied them to our single-cell RNA sequencing data. C #4 and #5 had a significantly higher expression level of ILC-specifying genes than other clusters, as shown by their relative expression in UMAPs (Fig. 2a). These higher module scores of C #4 and #5 were significant compared to the other clusters. Next, we divided the UMAP based on the tissue types to define the inter-tissue differences (Fig. 2b). We found LN contained the most cells in C #4 and #5 compared to BM or Spl. Then, we generated a module score using a list of genes with higher expression in NK cells than ILCs and applied it to our scRNA-seq data (Fig. 2c). Again, we found that cells with NK features were represented at a higher number in all the clusters except C #4 and #5. Analyzing the cells by tissue type demonstrated that both the BM and Spl contained proportionately higher NK cell gene features than the LN (Fig. 2d). Analyzes of percentages of the cells in each cluster in three tissues confirm the higher number of ILCs in the LNs (Fig. 2e). Further, LN contains more CD127⁺ cells than BM and Spl (Supplemental Fig. 1b, c).

Next, we analyzed the expression of known markers to authenticate the identity of C #4 and #5 as ILCs in LNs. We found a high level of *IL-7R* expression in C #4 and #5 (Fig. 2f), consistent with the notion that IL-7 is essential for the development and homeostasis of the ILCs, especially in the LNs[33,34]. In addition, these clusters possess a higher expression of *CD52*, which is expressed on ILCs from progenitor stages[35]. We also found the expression of *CD69*, a marker for tissue retention[36] (Supplemental Fig. 2a). These clusters also expressed *CD3D, CD3E*, and *CD3G*, which have previously been shown to be high in ILCs[31,37]. We generated a heatmap of 100 highly upregulated genes in C #4 and #5 and compared them with the remaining NK cell clusters (Supplemental Fig. 2a). We found transcripts encoding immediate early genes (IEGs), including *FOS, JUN, FOSB, AREG*, and *CD69*, were highly expressed in C #4 and #5. Importantly, transcripts related to cytotoxicity, including *GZMB, GZMA, GZMM, GZMH, GNLY, CTSW, CST7, NKG7*, and *PRF1* were expressed at much lower levels compared to NK cells (Supplemental Fig. 2a). Transcripts encoding several ribosomal proteins were expressed in C #4 and #5; however, their expression was shared by NK cells. Moreover, expression of NK cell-specific genes, including *TYROBP, FCGR3A*, and *KLRF1*, were comparatively low in C #4 and #5 (Supplemental Fig. 2a). Transcript levels of chemokines CCL3, CCL4, and CCL5, primarily produced by NK and T cells, were also lower in C #4 and #5 (Supplemental Fig. 2a).

To further validate that C #4 and #5 are ILCs, we compared the expression levels of TFs required to develop and maintain ILCs or NK cells (Fig. 2g). We found the expression levels of *GATA3* and *RORC* were higher, and *TBX21* and *EOMES* were lower in C #4 and #5 and the remaining clusters, respectively. However, the persistent expression of *TBX21* and *EOMES*, albeit at a lower level in C #4 and #5, indicated two possibilities.

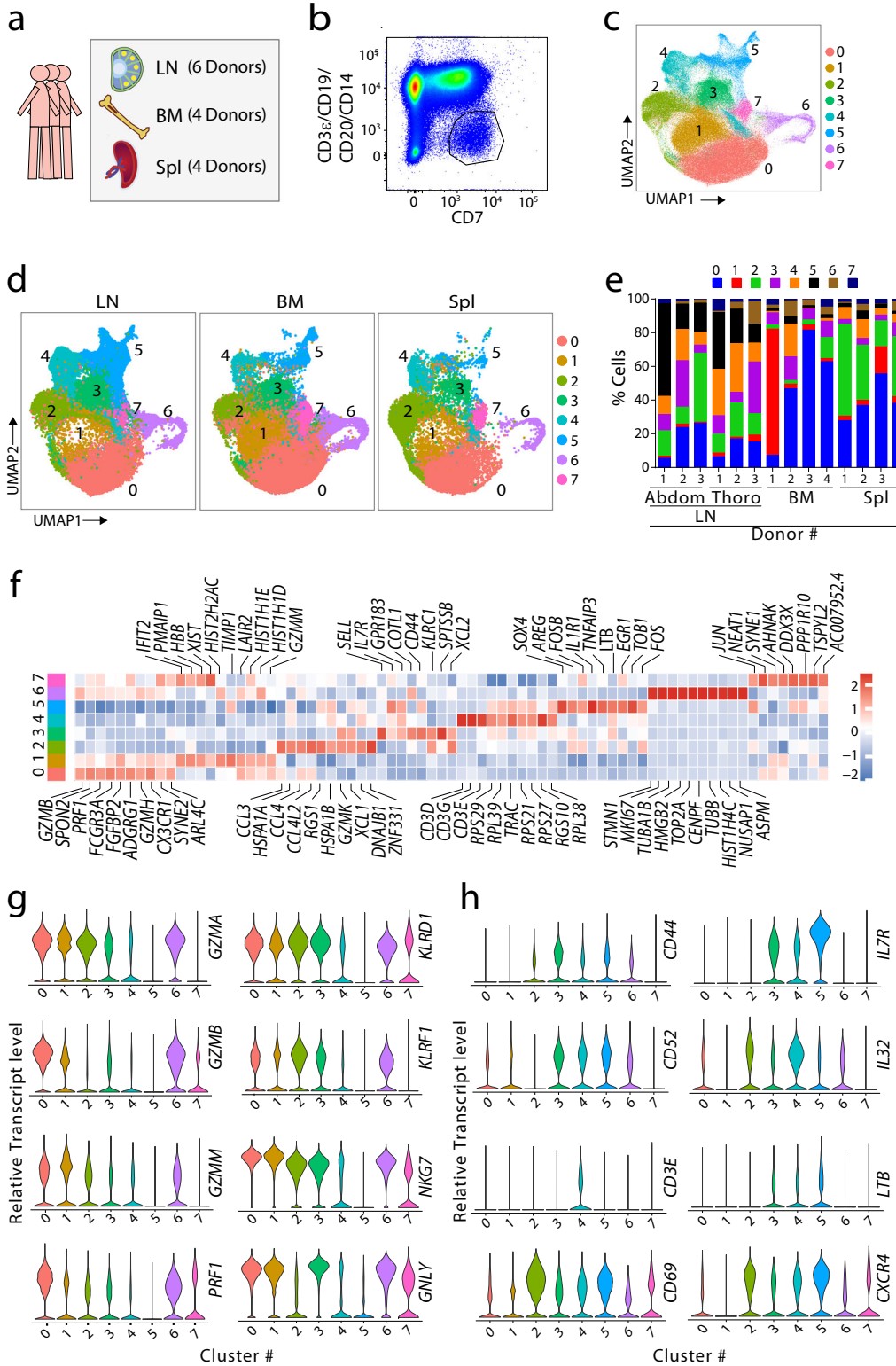

**Fig. 1 | scRNA-seq profiling of total ILCs in human bone marrow, lymph nodes, and spleen tissues. a** Graphical overview of study design. Human tissues were collected from individuals for scRNA-seq. Samples were obtained from 6 donors for LNs (lymph node), 4 donors for BMs (bone marrow), and 4 donors for Spls (spleen). **b** Sorting strategy for capturing the total ILC population. An anti-CD7 antibody was used for positive selection for ILCs. T cells, B cells, and monocytes were excluded by using anti-CD19, -CD20, -CD3ε, and -CD14 antibodies. **c** Identification of transcriptomically distinct ILC clusters. Clusters were defined by integration-based analysis of the differential expressed genes (DEGs) from all samples. Total cells clustered into 8 groups as depicted in a UMAP. **d** Dividing the UMAP into three tissues. The separated UMAPs show the distribution of the clusters in LN, BM, and Spl tissues. **e** The percentages of all ILCs in each donor separated by tissues, which include Abd (abdominal) LNs, Tho (thoracic) LNs, BMs, and Spls. **f** The top 10 DEGs in each cluster visualized by a heatmap. The color scale reflects a log2-fold change in Z-score. **g** The levels of expression are shown in violin plots for markers defining canonical NK cells. **h** Violin plots showing highly expressed genes in clusters 4 and 5.

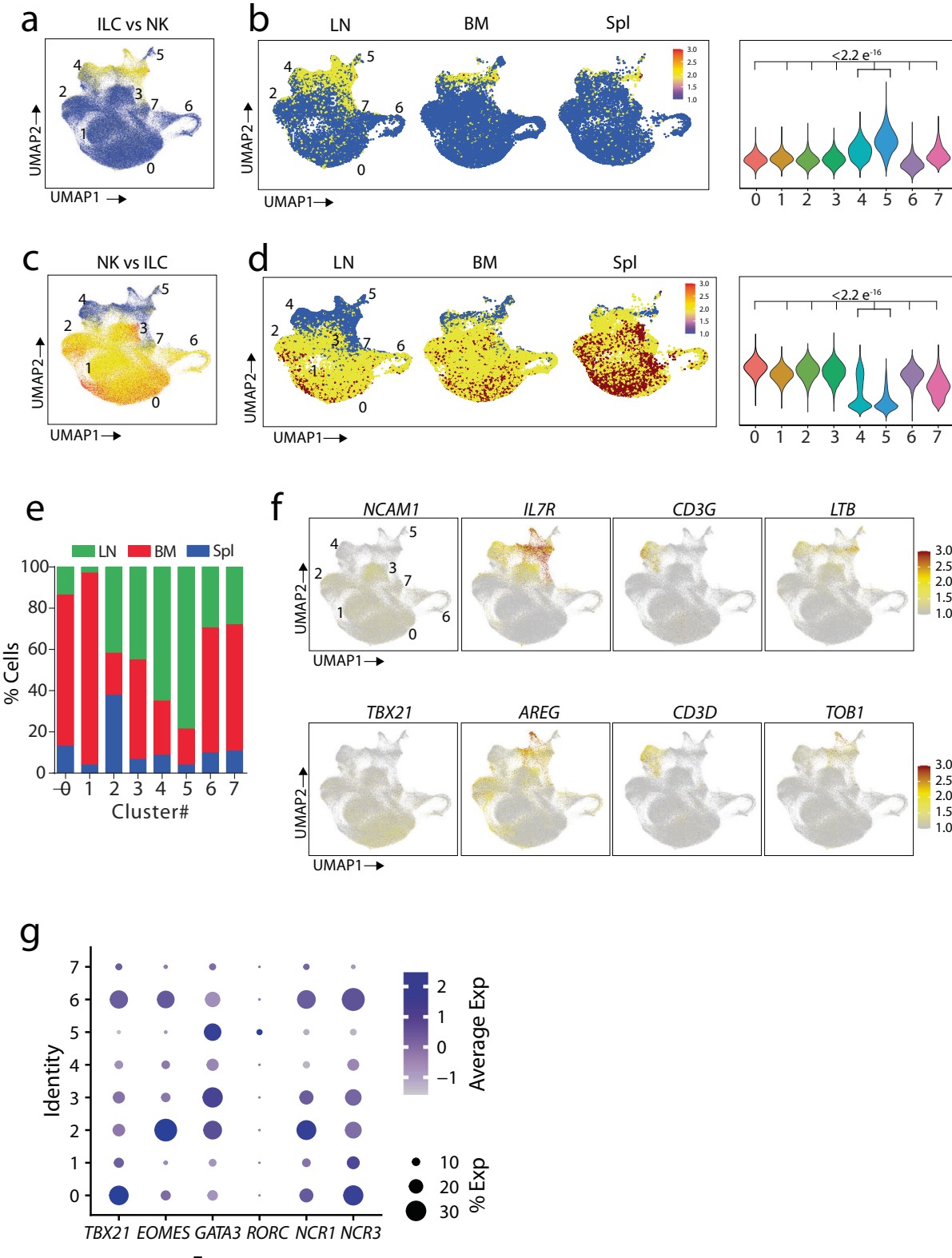

It can be due to the ILC1 subset, which shares a similar transcriptional profile of NK cells, or a smaller number of NK cells still present within clusters. Therefore, we reclustered C #4 and #5 and analyzed their transcriptomic profiles. We found eight subclusters (Supplemental Fig. 2b); among these, C #1 and #7 possessed transcripts indicative of NK features (Supplemental Fig. 2c). Application of a module score based on NK cell-specific gene signature validated that subclusters #1 and #7 are NK cells (Supplemental Fig. 2d). Therefore, we removed them from further analyzes. A second reclustering provided ten subclusters, and subcluster #8 contained NK features (Supplemental Fig. 2e, f). We confirmed this by module score analysis (Supplemental Fig. 2g) and removed it. The remaining nine clusters represented ILCs and were used for further studies. Collectively, we

**Fig. 2 | Human LNs contain a higher percentage of CD127⁺ ILCs compared to BM or Spl. a** Expression level of ILC gene list on the total ILCs. Module scores were calculated by the expression levels of gene lists specific for each cell type and visualized by UMAPs. Blue to red represent levels of expression of the gene list in each feature plot. **b** Dividing the UMAP into three tissues. The separated UMAPs show the distribution of the clusters in LN, BM, and Spl tissues. Expression levels of specific gene lists are shown in violin plots. P-values were calculated by Welch's t-test. **c** Expression level of canonical NK gene list on the total ILCs. Module scores were calculated by the expression levels of gene lists specific for each cell type and visualized by UMAPs. Blue to red represent levels of expression of the gene list in each feature plot. Expression levels of specific gene lists are shown in violin plots. P-values were calculated by Welch's t-test. **d** Dividing the UMAP into three tissues. The separated UMAPs show distribution of the clusters in LN, BM, and Spl tissues. **e** The feature plots show the expression of selected genes for helper ILCs on the combined LN, BM, and Spl tissues. Blue to red represent levels of expression of the selected genes. **f** Dotplots showing the expression of known markers for NK and ILCs. The size of the dots showed the percentage of the cells expressing genes, and the color indicates the significance of the expression. **g** The percentage of ILC clusters of LN, BM, and Spl tissues is shown by barplot. The tissues are color coded.

conclude that human LNs contain significant numbers of ILCs compared to the BM or Spl. By contrast, NK cells are predominantly present in the BM and Spl compared to the LNs.

### Unique transcriptional signatures of ILCs in human LN

To identify the individual ILC subtypes, we used the gene signatures of ILC1, ILC2, ILC3, and pILCs[31]. We generated gene sets containing a minimum of 100 genes for each ILC subtype and applied that to our single-cell RNA sequencing data from all three tissues in module scores overlayed onto a combined UMAP[38]. Using these, we defined all three ILCs. C #0 and #1 were identified as ILC1, which was validated by a significant increase in the module score. The module score for ILC2 was highest in C #9, which had a limited cell number. C #2, #3, #5, and #6 were defined as ILC3 (Fig. 3a). Although C #9 expressed several genes that were shared between ILC1 and ILC2, a significant module score confirmed that it represents ILC2. Similarly, C #4 expressed a shared transcriptomic profile with ILC3; however, it contained significantly higher module score values for HSC gene features. We generated a gene list of 20 known CLPs to validate and applied it to our data. We found a majority of the cells in C #4 were highly positive for CLP gene features. It is important to note that C #4 was predominantly present in the BM compared to LN or Spl. In contrast, C #7 was only present in the LNs and formed a distinct cluster away from others. Previous studies in the human colon identified a subset of ILCs that lacked the master TFs needed for ILC development[32]. However, these cells had the potential to become mature ILCs and therefore defined as naïve ILCs. We generated a gene list for nILCs from this study and applied it to our data. C #7 showed the highest expression of gene signatures associated with nLCs (Fig. 3a).

Next, we generated a heatmap of ten highly expressed DEGs for each ILC and the CLP clusters (Fig. 3b). C #4 expressed high levels of HSC-defining genes, including *CD34, CD38, SOX4, ITGA4, and CD74* (Figs. 3b, c). ILC1 clusters expressed *CD3E, CD3D, CD3G*, and *CCR7*. The ILC2 cluster contained high levels of *S100A4, S100A6, HPGDS, PTGDR2*, and *IL32*[39]. ILC3 clusters have higher expression of *CD300LF, NCR2, IL23R*, and *IL1R1* (Fig. 3c). The expression of known TFs further validated that C #0 and #1 are ILC1 (*TBX21*); C #9 is ILC2 (*GATA3*); and C #2, #3, #5, and #6 are ILC3 (*AHR*) (Fig. 3c). Next, we performed flow cytometric analyzes to confirm the presence of non-NK ILCs in human LNs and BM. Antibodies against CD3ε, CD19, CD20, and CD14 were used to exclude T cells, B cells, and monocytes[30]. The remaining CD7⁺CD3ε⁻CD19⁻CD20⁻CD14⁻ population was divided into CD127⁺ and CD127⁻. The majority of the CD7⁺ cells (63.8%) were CD127⁺. We selected the CD127⁺ cells and analyzed them for CRTH2 (PTGDR2) and CD117 (c-KIT) positivity. In line with our transcriptomic data, we found abundant ILC3s (CD7⁺CTRH2⁻cKIT⁺) and only minimal ILC2s (1.98%, CD7⁺CTRH2⁺cKIT⁻) in the LNs (Fig. 3d). CD7⁺CRTH2⁻cKIT⁻ cells represented ILC1 and immature CD56^Bright NK cells (35.1%). Similar analyzes of human BM showed only a smaller number of CD7⁺ cells, which were CD127⁺ (8.72%), and ILC2 and ILC3 were equally present (Fig. 3e, Supplemental Fig. 3a, b).

To determine the gene regulatory network (GRN) operating in individual subtypes of ILCs, we employed a single-cell regulatory network inference and clustering (SCENIC) analysis (Fig. 3f). Each TF regulates multiple target genes, which are grouped into units defined as regulons.

We defined target genes regulated by a TF identified from motif enrichment analyzes using the GRNboost2 fast GRN algorithm[40]. The ranked distribution of AUCell scores across individual cells from a binary output determined the threshold for active and inactive regulons.

LEF1 and TCF7 regulons are predominantly operative in ILC1. In addition, LEF1 is involved in the direct induction of *IL7R* (CD127, IL-7Rα) and c-Myc[41]. Consistent with this data, we observed *MYC* and *LEF1* regulon activity and their gene expression (Supplemental Fig. 3c). The regulon activity also indicates high expression of gene sets related to STAT3, ETS1, IKZF, RUNX1, and FOXO1 regulons. The gene regulatory network analysis for ILC2 revealed that this cluster had upregulated GATA3 regulon and BCL3, IRF3, CTCF, TAF7, and BATF regulons as are necessary for ILC2 development (Fig. 3f). Regulon activity and the gene expression of TFs, including *RUNX2, AHR, EGR1*, and *BHLHE40*, are higher in ILC3 subsets than other ILCs (Supplemental Fig. 3d). Motif analysis for the promoters of the genes indicated specific binding sites for these TFs (Supplemental Fig. 3e). ILC3 subset in the LNs expresses Activating Transcription Factor (ATF) family, including *ATF1, ATF2, ATF3, ATF4*, and *ATF5*. ILC3 also expressed high levels of other activation markers, including *JUN, JUND*, and *FOS*, that form AP-1 heterodimers. Collectively, these data indicate the presence of three main subtypes of the ILCs in human LN with specific gene signatures and GRNs.

### Development of ILC progenitors in the BM

Next, to identify the early developmental stages of ILCs in human BM, we utilized the same clustering and the module scores of gene signatures employed in Fig. 3a for ILC1, ILC2, and ILC3. It is important to note that although we removed the lineage-committed myeloid (CD14), T (CD3ε), and B (CD19/CD20) cells by positive selection during cell sorting, the remaining CD7⁺ population contained HSCs, progenitors, ILCs, and NK cells. For CLPs and pILCs, we used *CD34, SOX4, STMN, CD38* and *NFIL3, ID2, ZBTB16* genes, respectively. Using these gene signatures, we identified cells representing HSCs/CLPs, nILCs, ILC1, ILC2, and ILC3 in the BM (Fig. 4a). Among the 10 clusters, the majority of the cells in the BM were present in C #4, and other clusters contained only minimal cell numbers. C #4 possessed eighty percent of the cells in the BM expressing CLP/Early innate lymphoid progenitors (EILPs)/pILCs gene features. (Fig. 4b). Consequently, we found fewer cells expressing ILC1, ILC2, or ILC3 gene features within C #4 and in the BM itself. Among the tissues, as expected, BM contained the highest number of cells expressing HSC/CLP gene features compared to Spl and LNs (Fig. 4c). To delineate the developmental stages, we generated the Monocle-2-based trajectories overlaid on UMAPs defining individual clusters, which indicated an orderly transition of CLPs into EILPs/pILCs to ILC3 and ILC1 (Fig. 4d). Cells in C #4 started the pseudotime trajectory, indicating the location of HSCs/CLPs, EILPs, and pILCs. Utilization of features defining CLPs and early progenitors further confirmed this unique transition pattern based on gene expression levels (Fig. 4d).

Next, we analyzed gene expressions related to early developmental stages of ILCs by overlaying feature plots onto UMAPs. Expression of *CD34* defined HSCs and CLPs. Earlier reports have shown that the first uncommitted ILC progenitor potential resides within the α4β7-positive fraction of the CLPs[42]. We found the expression of *ITGA4* transcripts in a largely

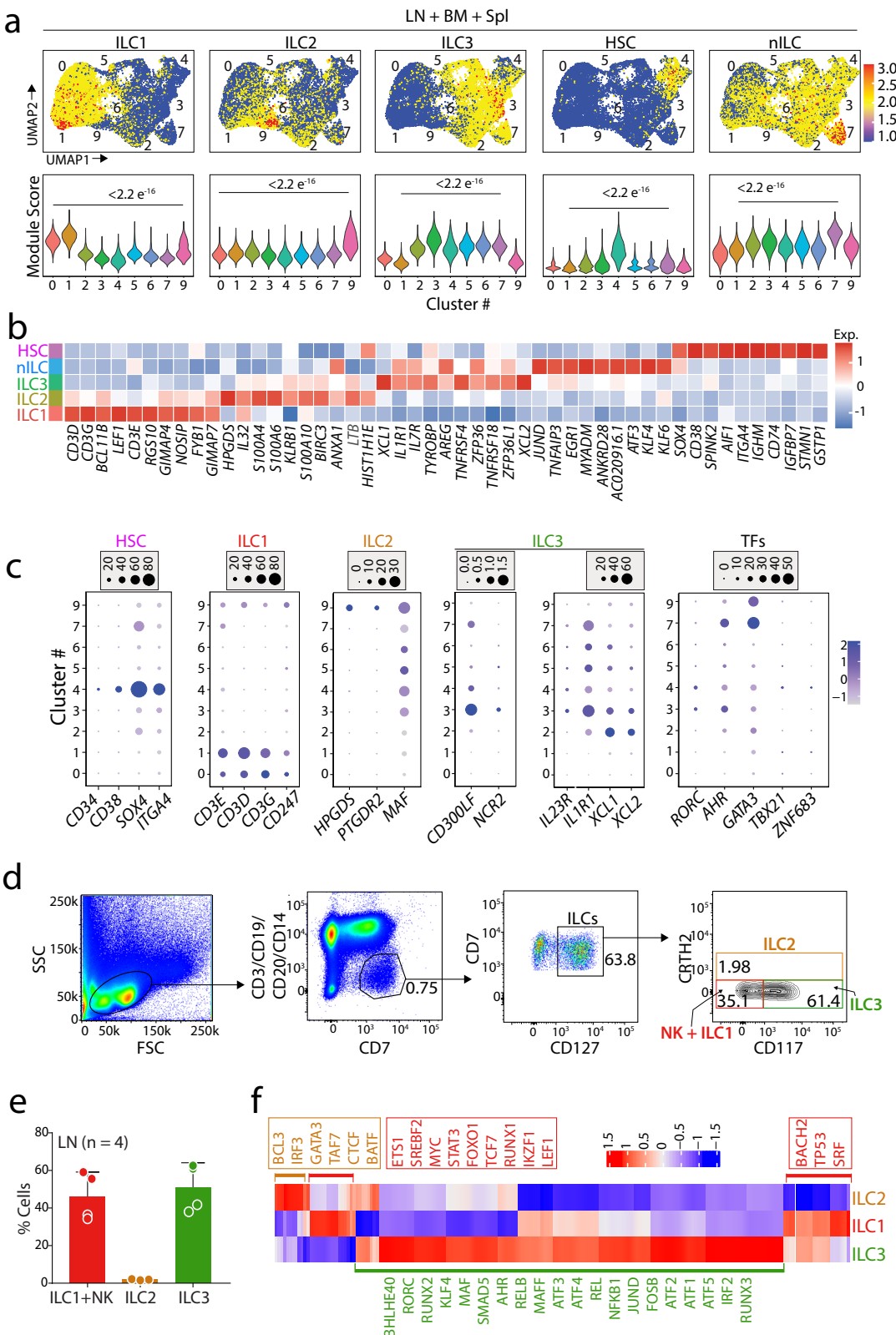

non-overlapping cell population that expressed lower or no *CD34* transcripts. This implies the human BM contains the CD34$^-$α4β7$^+$ progenitor with ILC potentials. *SOX4* is predominantly expressed in C #4 (Fig. 4e), defining these cells with the highest stemness. Transcripts encoding MEIS1 that protect and preserve HSCs by restricting oxidative metabolism were moderately present in C #4[43]. The increased expression of *NFIL3* coincided

with *SOX4* and *CD38* (Fig. 4e), marking the emergence of CLPs[44]. Earlier studies have shown that the lack of Nfil3 abolished the development of all subsets of ILCs[45,46]. Our data demonstrates the expression of *NFIL3* transcripts in a subset of cells early in the pseudotime trajectory. Early innate lymphoid progenitors (EILPs) are a population of multipotent progenitor intermediates between all lymphoid progenitors (ALPs) and progenitor

**Fig. 3 | Identification of ILC subsets in human lymph nodes. a** Identification of ILC subsets in combined BM, LN, and Spl tissues. Module scores were calculated by the expression levels of gene lists specific for ILC1-3, HSC/CLP (hematopoietic stem cell/common lymphoid progenitor) and nILC (naïve ILC) and visualized by UMAPs. Blue to red represent levels of expression of the gene list in each feature plot. Expression levels of specific gene lists are shown in violin plots. *P*-values were calculated by Welch's t-test. **b** Top 10 DEGs in ILC subtypes and ILC progenitors visualized by heatmap. The color scale reflects Z-score of Log₂ fold change. **c** Dot plots showing expression of known markers for subsets of mature ILCs, progenitors and transcription factors (TF). The size of the dots shows the percentage of cells

expressing genes, and the color indicates significance of the expression. **d** Identification of ILC subtypes in the LN tissues. An anti-CD7 antibody was used to capture all ILCs, including NK cells. T cells, B cells, and monocytes were excluded by using anti-CD19, -CD20, -CD3ε, and -CD14 antibodies. CD7⁺CD127⁺ cells were identified as helper ILCs. Anti-CD117 and -CRTH2 antibodies were used to identify ILC1, ILC2, and ILC3 populations. **e** The percentage of subtypes in total ILCs with each sample noted as a dot. Error bars indicate the standard diviations with mean values. **f** A heatmap showing regulon activities in each ILC subtype. Blue to red indicates the level of activity for specific regulons.

---

ILCs (pILCs)[47]. Previous findings have shown that TOX is required to convert ALPs to EILPs[48]. We found the presence of transcripts encoding TOX in a moderate number of cells that expressed both *NFIL3* and *IL7R* transcripts (Fig. 4e). Expression of *IL7R* is higher in a population of cells that are low in *NFIL3* and *TOX* transcripts. This may be because the mature ILCs, particularly ILC3, express much higher levels of IL-7R. In murine models, a common helper ILC progenitor (CHILP) that expresses *Zbtb16* (Plzf) among the pILC fraction has been described[4,5]. Therefore, we next analyzed the expression of *ZBTB16*. We found the presence of *ZBTB16* transcripts in subsets of cells that are either *IL7R* transcript positive or negative. IL-15R-mediated activation promotes the expression of NFIL3, which is required for the transcription of ID2 and GATA3 in EILPs. We found both the expression of ID2 and GATA3 transcripts in a subset of cells away from the CLPs or CLPs with ILC potentials. These results conclude that the human BM contains distinct developmental stages of ILCs, including CLPs (or ALPs), non-committed CLPs with ILC potentials, EILPs, and pILCs.

We performed SCENIC analysis on the BM cells to identify the specific TFs responsible for each stage of the ILC development (Fig. 4f, g). GRN analyzes confirmed the role of SOX4 regulon as its expression level is high in cells at the start of the pseudotime trajectory. SOX4 expression and its regulon activity decreased in cells present at the later stages of the pseudotime trajectory, implying an active maturation process (Fig. 2f). *BCL11A* is essential for the survival of CLPs and lymphopoiesis[49]. We found that the expression level of the *BCL11A* and its regulon activity gene is high in the CLPs and steadily decreases as cells mature. *IRF8* is expressed at a high level, and its regulon is coactivated with MEF2C, RUNX2, and SP1B regulons (Fig. 4F). The developmental progression from pILCs to mature ILCs was also seen in the BM as determined by RORC, AHR, KLF7, and TCF7 regulons (Figs. 4f, g). Next, we analyzed the three major clusters (C #1, #3, and #4) present in the BM for active regulons (Fig. 4g). Five regulons, SAP30, IKZF1, ZNF362, SPI1, and FOXJ3, were dominantly active in all three clusters. ZNF362 and SPI1 were universally present in all the cells (Fig. 4g-i). Three additional sets of regulons were expressed in C #4. These included KLF9, CEBPB, PRDM1, TBX21, and RUNX3 in the first group (Fig. 4g-ii), BCL11A, MEF2C, RUNX2, SPIB, and IRF8 in second group (Fig. 4g-iii), and ZNF22, TFDP1, EBF1, MYB, and LYL1 in the third group (Fig. 4g-iv). Cluster #3 possessed BHLHE40, TCF7, RORC, and AHR regulons, indicating the presence of committed ILC3 in the BM (Fig. 4g-v).

As BM and LN contain C #4 as the progenitor population, we compared the transcriptomic differences between the tissues using DEGs (Supplemental Fig. 4a). In the BM cells, C #4 expressed high levels of genes and regulons indicative of early developmental stages, including *ITGA4*, *CD38*, *CD34*, *RUNX1*, *RUNX3*, *MYB1*, and *ZEB2*. Conversely, LN C #4 cells expressed genes defining committed pILCs, including *KIT*, *ID2*, *IL1R1*, and *IL7R*. It is important to note that some of the highly expressed genes in in C #4 of BM were shared with C #4 cells of LN. However, most of the genes that are highly expressed in LN were not shared with C #4 of BM (Supplemental Fig. 4a), potentially suggesting a link between the ILC progenitors in BM and differentiating ILCs in LN. Next, we identified the gene regulatory networks operating in the LN and BM. The regulon activities of C #4 in LN and BM reveal three active groups. The first group, including AHR, ATF, and NFIL3, is common between the two tissues. In the previous section, we identified the expression of NFIL3 in EILP as a transient stage. The next group, mostly

active in BM cells, expressed TBX21 and RUNX1 regulons and has been previously defined as an early developmental stage[50]. Finally, the last group of regulons is mostly active in LN cells. These regulons mostly consist of genes known for later development stages in mature ILCs. This includes RORC, RUNX2, BHLHE40, and the ATF family. (Supplemental Fig. 4b). We analyzed major signaling pathways using Gene Set Enrichment Analyzes (GSEA) using Reactome and Msig Data Base (MsigDB, Hallmark gene sets) databases to differentiate the pathways that operate in C #4 from BM and LN (Supplemental Fig. 4c). Cell-cycle signaling was significantly enriched in BM compared to LN. In contrast, the TLR pathways, including TLR3 cascade and MyD88 independent TLR4-mediated activation, were enriched in the LN C #4. Together, our single-cell data indicate the developmental progression of ILCs begins in the BM and continues in the LN via nILC progenitors.

## ILC progenitors develop in the BM and commit to naïve ILCs in the LNs

We identified C #7 as naïve ILCs exclusively present in the LN tissue, raising the question of whether there is a direct link between progenitor ILCs in the BM and naïve ILCs in the LNs. First, we defined the differentially expressed genes between C #4 of BM and C #7 of LN using heatmaps (Fig. 5a). BM C #4 highly expressed *SOX4*, *TOX2*, *CD38*, and *BCL11A* genes, which are progenitor markers, and cell cycling genes such as *CDK6* and *STMN-1*, that function as an important regulatory protein of microtubule dynamics and play an important role in cell-cycle progression and cell migration[51]. We subset these two and reclustered them to gain insights into the developmental progression of ILCs from the BM to the LN. We identified five distinct clusters. This unsupervised clustering identified LN naïve ILCs as a separate cluster from BM cells (Fig. 5b). To assess the developmental progression from BM progenitors to naïve ILCs, we generated a pseudotime trajectory[52] using a Monocle3 (Fig. 5c). The Pseudotime and developmental progression uncover the link of the BM progenitor to nILCs and the possibility of early development of ILC in the LNs. Computational ordering of cells in an unsupervised manner using maximal transcriptional similarity between successive pairs of cells allowed us to define the transcriptomic continuity of the cells. Pseudotime ordering formed a gradual progression from BM HSCs to naïve ILCs. The starting point of the pseudotime trajectory is HSCs expressing *CD34*, *TCF7*, *SOX4*, and *ITGA4*, followed by two major branch points (BPs), indicating two distinct cell fate (C_F) decisions. Cells in C_F1 progress to become mature ILCs in BM, and cells falling in C_F2 develop into naïve ILCs. Expression of individual genes through the pseudotime trajectory indicates specific requirements of the naïve ILC development (Fig. 5d). The genes, including CD34, SOX4, and NFIL3, are expressed solely in the BM cells. GATA3, TCF7, and ZBTB16 expression starts from BM cells and increases through the development of the naïve ILC cluster. Expression of *IL7R*, the canonical maker of ILCs, begins from the later part of the BM trajectory and peaks in all the naïve ILCs, indicating the importance of this receptor in the ILC development.

We performed SCENIC analysis to define the global transcriptional changes orchestrated by the GRN. Using this analysis, we identified 76 co-regulated regulons further classified into five active groups (Fig. 5e, f). Regulons from BM cells were clustered into four groups. The first set contained regulons operated by BCL11A, SOX4, BCLAF1, MAF, MEF2C,

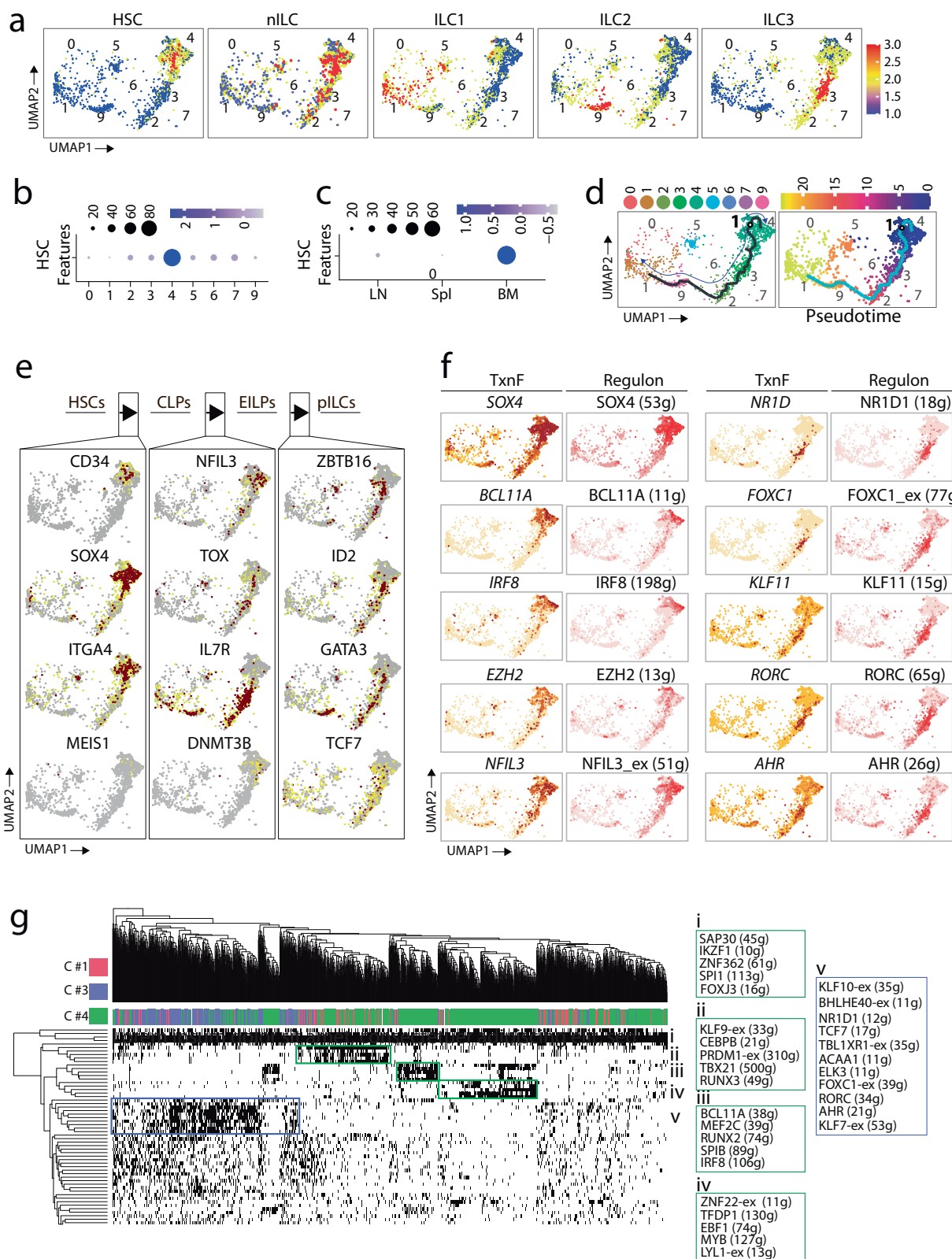

and SMARCA4, predominantly active in most BM cells. The next major group is TBX21, RUNX1, RUNX3, and MYB, which also operate in the BM cells. The next two smaller regulons were operated by (ETS2, NFIL3) and (TCF7, RORA). On the contrary, the active regulons in the naïve ILCs are related to immediate early genes (EGR1, FOS, JUNB, JUND) NFKB family (NFKB1, NFKB2, REL, RELB), which indicates the role of NF-κB pathway

in ILCs development. The activity of individual regulons and the expression of TFs overlay on the UMAP validated the role of regulons. NFIL3 regulon is more active in the BM cells, while the expression of the *NFIL3* gene continues in naïve ILCs. TBX21 regulon and its gene expression were more obvious in the mature ILCs as the role of this gene has been identified in ILC1 development[5]. *STAT4, GATA3,* and *RORC* expression and regulon

**Fig. 4 | Human ILC development initiates from bone marrow. a** Identification of ILC subtypes in BM. Module scores were calculated using expression levels of gene lists specific to each cell type and visualized by UMAPs. Blue to red represent expression levels of gene lists in each feature plot. P-values were calculated by Welch's t-test. **b** Expression of HSC/CLP gene features per cluster displayed as dot plots. The size of each dot shows the percentage of the cells expressing the noted gene, with color indicating the significance of expression. **c** Dot plot displaying expression of HSC/CLP features per LN, BM, and Spl. **d** Pseudotime analysis of the progenitor cluster to the mature ILCs using the Monocle3. The line indicates the direction of developmental progression from progenitors to mature ILCs. UMAPs display clusters (left), and color represents the temporal dimension of the cells in each stage (right). **e** Schematic figure showing stages for early development of ILC in BM. The arrows indicate transient expression of TFs in each HSCs, CLP (common lymphoid progenitor), EILP (early ILC progenitor), and PILC (progenitor ILC) stage. **f** The expression of TFs and their related regulons in the BM clusters. The darker color represents a higher expression of TF or higher regulon activity. **g** Regulon activity in C#4 (ILC progenitor) and two mature clusters C#1 and C#3 The color bar on the top represents each cluster and related high activity regulons. The regulons are grouped in the color boxes based on the same pattern of activity.

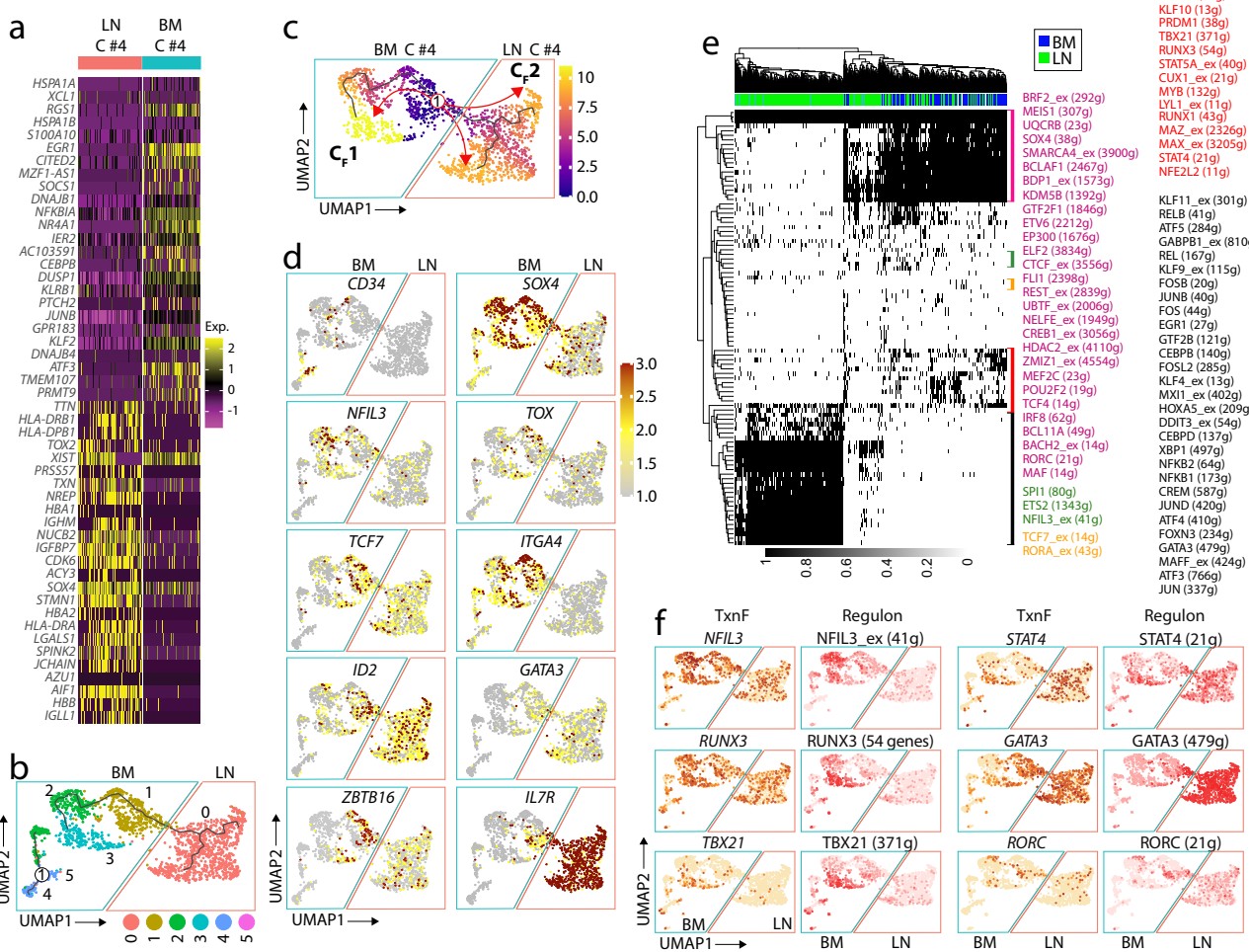

**Fig. 5 | Developmental progression in the LNs indicates a naïve population committed from early progenitors. a** Heatmaps of the top significant DEGs in C#4 of the BM and C#7 of the LNs. Rows and columns represent genes and cells, respectively. Cells from the same cluster are grouped together. The color scale reflected Z-Score from log2 fold change. **b** UMAP plot showing the clustering of combined BM progenitors and LN naïve ILC clusters. The reclustering identified 6 distinct clusters. Tissues are color-coded (LN blue; BM red). (1) indicates the starting point of the pseudotime trajectory on the separated partition (left side) of the cells. **c** Pseudotime trajectory analysis of BM progenitors and LN naïve ILC clusters. The line indicates developmental progression, and the colors represent the temporal dimension of cells in each stage. **d** Expression levels of genes related to the early progenitor stage to more committed ILCs shown in feature plots. Darker red represents higher expression levels. **e** Regulon activities in the BM progenitors and LN C #7 showing as a heatmap. The color bar represents each tissue, and the gray scale indicates regulon activity levels. The regulons are grouped in the color boxes based on their pattern of expression. **f** Expression of TFs and related regulons in BM progenitors and LN C #7. The darker color represents a higher expression of TF or higher activity of regulon.

activity begin in the pILCs and continue to naïve ILCs. Collectively, these data define the transcriptional profile of the early developmental stages of ILCs in the BM and LNs.

To further define the differences in the profile of naïve ILCs to C #4 of BM, we analyzed major signaling pathways using GSEA. Similar to what we found in comparing the C #4 in LN and BM, TLR pathways are significantly enriched in naïve LNs. In addition, the signaling pathways related to major cytokine pathways, including IL-4 and IL-13, were also enriched in LNs. On the contrary, the major upregulated pathways in the C #4 in the BM were related to proliferation, including G1/S transition and cyclin-related pathways, as shown in (Supplemental Fig. 5a). In the previous section, we have shown the location of HSC/CLP/EILP and pILCs in C# 4 using their specific gene expression. We isolated and reanalyzed these subsets to validate the link between the progenitor

cluster and nILC. The results indicate the continuation of pILCs at the end of C #4 to nILCs in the LNs. To understand the potential relationships between the nILCs and more mature ILC subtypes, we performed trajectory analysis via Monocle. Overlaying these trajectories with the ILC clusters indicates the direct link of nILC to these subsets. Generally, using these methods, we could show the direct developmental stages from pILC to naïve ILCs and to the ILC1 and ILC3 clusters in the LNs (Supplemental Fig. 5b, c).

We generated heatmaps representing highly expressed genes to determine the similarities in gene expression between the abdominal and thoracic LN ILCs compared to the Spl or BM (Supplemental Fig. 6a). We found a higher expression of similar genes between ILCs from abdominal and thoracic LNs. While ILCs from the Spl, another secondary lymphoid organ, expressed a unique set of genes, they shared several genes expressed in the LN ILCs. However, ILCs from the BM shared only a minimal number of genes with ILCs from the LNs, expressing a unique set of genes (Supplemental Fig. 6a). Next, we identified the differentially expressed genes between the abdominal and thoracic LN ILCs. We found several transcripts encoding transcription factors KLF2, KLF6, JUND, FOS, JUN, NFKBIA, FOSB, and GATA3 in ILCs from the abdominal LNs (Supplemental Fig. 6b). In contrast, the transcripts encoding heatshock proteins (HSPA1B,

HSPA1A, DNAJB1, and HSPE1) and ribosomal proteins (RPS29, RPL37, RPL36, RPL38, and RPL22) were predominantly expressed in ILCs from thoracic LNs.

## Transcriptionally distinct subpopulations of ILC1 and ILC3 in human LNs

We found the highest percentage of ILCs in the LN compared to BM and Spl. To gain further insights into the transcriptomic profiles of ILCs in the human LNs, we reclustered ILC1 and ILC3. Since ILC2 was represented only by one cluster, we did not further subdivide it. We identified three distinct sub-clusters in the ILC1 population (Fig. 6a). The UMAP plot and the cell number in three tissues indicate a higher number of all three ILC1 sub-clusters In LNs than BMs and Spl (Fig. 6b). C #1 is the most abundant cluster in the BM, while all three sub-clusters were found in Spl, albeit in much fewer numbers. To identify the transcriptomic differences of ILC1 in LNs, BMs, and Spl, we compared the gene signature of total ILC1 in these tissues (Fig. 6c). We uncovered transcripts encoding the subunits of the CD3 signaling complex (*CD3E, CD3D, CD3G*) that are commonly expressed on ILC1 in all three tissues. We also found *CD48* expression in the common genes of ILC1 in three tissues. CD48 associates with CD2 and efficiently brings the Src family protein kinase LCK and LAT to the

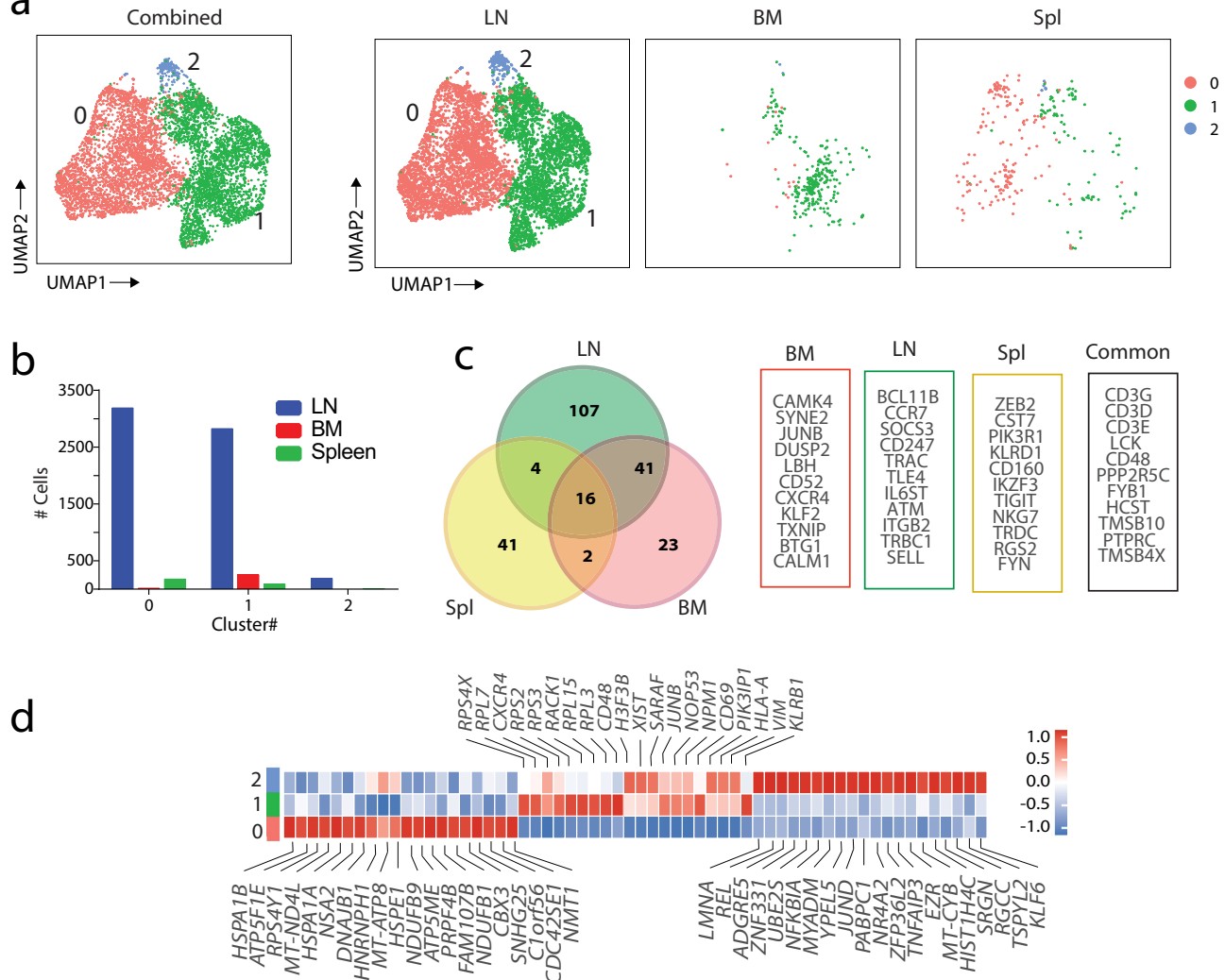

**Fig. 6 | ILC1 in the human LNs contains three distinct clusters. a** The ILC1 reclustering defined three transcriptomically- distinct clusters. UMAP visualization of ILC1 clusters color-coded based on tissue origin. **b** The barplot showing the actual number of each cluster. The LNs have higher total ILC1 compared to BMs and Spls.

Cluster #1 is higher in the BM samples. **c** The unique and common genes for ILC1 in LN, BM, and Spl tissues. The colors indicate each tissue and the black box indicate the common genes for all three tissues. **d** The heatmap showing top DEGs in each of the three ILC1 clusters. The color scale reflects Z-score from log2 fold change.

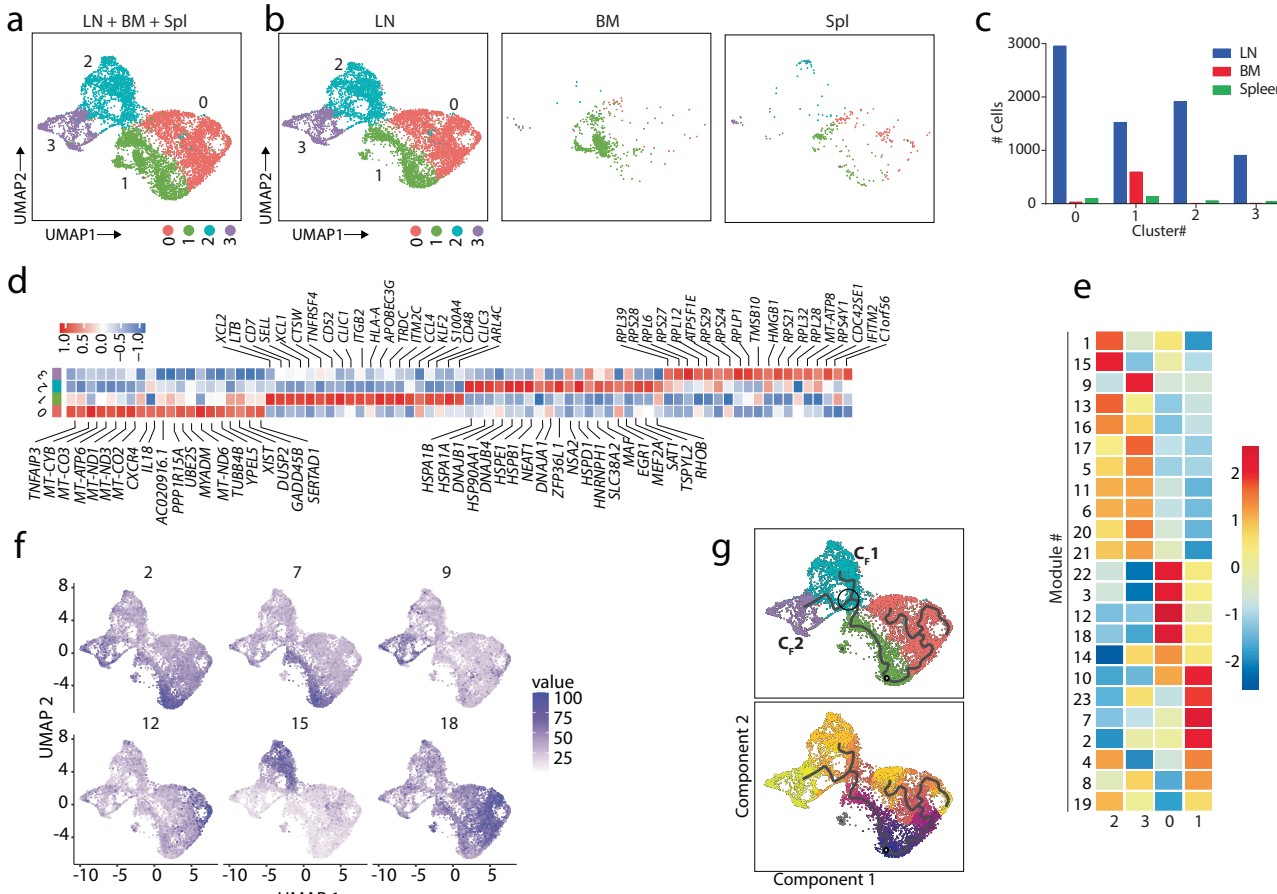

**Fig. 7 | Identification of four transcriptomically- distinct clusters in human ILC3 in the LNs. a** The ILC3 reclustering defined four transcriptomically- distinct clusters. Each color in the UMAP plot represents one cluster in the ILC3 population. **b** UMAP visualization of ILC3 clusters color-coded on the basis of tissue origin. **c** The barplot showing the actual number of each cluster in the tissues. The LN has a higher number of total ILC3 compared to BMs and Spls. Cluster #1 is higher in the BM samples. **d** The heatmap showing the top DEGs in each of the four ILC3 clusters.

The color scale reflects Z-score from log2 fold change. (**e**) & (**f**) Defining the active gene module in each ILC3 subset. The level of expressed genes in each module is shown as a heatmap (**e**) and as a feature plot for selected modules (**f**). **g** Pseudotime analysis on the ILC3 subsets. The pseudotime axis indicates the temporal changes of expression of the genes starting from C#1. The black circle indicates the branch point to C#2 for cell fate 1 and C#3 for cell fate 2.

TCR/CD3 complex[53]. High expression of *LCK* in all ILC1 subsets could suggest the possible activation of the signaling pathway associated with the Src family.

In addition to the common genes for three tissues, some features are highly expressed only in LNs, BMs, or Spl. Most migratory ILCs are ILC1s, and they could enter LN directly from the circulation using CD62L and CCR7[54]. LN-specific genes of ILC1 show a high level of *CCR7* and CD62L (*SELL*) in the LN ILC1, indicating the migratory characteristic of these cells. ILCs have an important role in producing different cytokines. IL-6 can induce SOCS3 expression through STAT3 signaling[55]. We also found that IL-6 family signal transducer (IL6ST) and SOCS3 genes are highly expressed in LN ILC1. Next, we analyzed ILC genes highly expressed in BMs and Spl. For BM, the specific genes include CD52, cytokine receptors *CXCR4*, and TF (*KLF2, JUNB*). The Spl ILC1 has more NK cell features. They express *CST7, NKG7, KLRD1*, and *CD160*, which are involved in the cytotoxic function of NK cells. These analyzes indicate the common genes and unique transcriptomic signature of ILC1 for LNs, BMs, and Spl.

Next, to identify the subset-specific gene expression and their signaling pathways, we generated a heatmap of DEGs showing the level of expression and significance of the top-regulated genes in three ILC1 clusters (Fig. 6d). These analyzes revealed that individual sub-clusters in an ILC population harbor highly variable transcriptomes[31]. C #0 highly expressed transcripts of heat shock proteins (*HSPA1B, HSPA1A, DNAJB1, HSPE1*). In addition, pathway analysis of the highly variable expressed genes found the HSP90

chaperone cycle for steroid hormone receptors (SHR) as the most significant pathway in this cluster. C #1 has an activated condition with high expression of ribosomal proteins, CD48, and KLRB1. This cluster expresses the activation marker CD69. These data could suggest C #1 as the potential cluster newly arrived from the BM to LN. Interestingly, among the three ILC1 clusters, C #1 is the most prevalent in BM, consistent with identifying cluster #1 as a possible cluster newly coming from BM to LN. The top DEGs in C #2 are the most distinctive set, while this cluster has some genes that overlap with C #1 (*JUNB, NOP53, SARAF*) (Fig. 6d). C #2 possesses a unique gene profile and shows more regulatory function by expressing *REL, NFKBIA, TNFAIP3, and KLF6* genes.

After defining subsets for ILC1, we performed unbiased clustering on the subsets of the ILC3 and regrouping them to have further insights into the transcriptomic profiles of ILC3 in the human LNs PCA analysis identifies four different subsets for ILC3 in the LNs (Fig. 7a). Separating the ILC3-subsets in three tissues shows all four subsets exist in the LN with the highest number of cells related to C #0 and #1 (Fig. 7b, c). BM cells were highly enriched in C #1. Spleens possess all subsets with a lower number of cells than LNs. (Fig. 7b, c). A hierarchical heatmap of the top 20 DEGs from all the unsupervised ILC3 clusters in the LN validates the distinct transcriptomic profile of the ILC3 subset (Fig. 7d). C #0 is the most abundant cluster in the LNs. It contains activated cells with high expression of immediate early genes, including *CXCR4, JUND, DUSP2*, and *GADD45B*. This cluster also expresses a high level of IL-18-encoding transcripts and

mitochondrial genes. C #1 of ILC3 shows a higher level of the genes involved in the early development of ILCs, including *SELL, CD7, XCL1, XCL2,* and *LTB*. The presence of this cluster as the major cluster in BM validates this cluster as an early-stage committed ILC3s that could be newly arrived from BM. Cells in C #2 expressed high levels of transcripts encoding heat-shock proteins (*HSP1A1, HSPB1, HSPD1, HSPH1, HSPE1*). The ILC1 subcluster and ILC3 clusters both have a high expression of heat shock protein. That could be an indication of activated ILC cells in these two groups. C#3 is the high ribosomal cluster because it shows high expression of genes related to ribosomal proteins and translation. Finally, C #4 in ILC3 has a considerable expression of transcripts for ribosomal protein in addition to a distinct expression of *IFITM2* and *C1orf56* genes. The clustering of ILC3 in LNs indicates the heterogeneity of them in the LNs with the unique transcriptomic signature for the subsets.

Next, to identify the potential spectrum of activation states of these ILC3 subsets, we performed pseudotime trajectory analysis (Fig. 7e–g). After finding the DEGs for all the clusters, we used Monocle3 to generate gene modules overlayed onto the UMAP. The genes involved in modules are shown (Supplemental Fig. 7a). Modules #2, 7, 9, 12, 15, and 18 are active modules in ILC3 clusters, as shown by heatmaps (Fig. 7e). The expression of each module is shown on each cluster (Fig. 7f, Supplemental Fig. 7a). Module #12 and 18 genes are highly expressed in C #0 and include *HES4*, *DUSP2*, and *TOB1*. Modules #2 and 7 are exclusively high in C #1. Identified early genes include *XCL1* and *XCL2*. These data validate our finding that C #1 is the earlier stage of ILC3 subsets. Other genes that could be found in this module include *SOCS3* and *KLRC1*. Module #15 is highly expressed on C #2 and contains transcripts encoding HSPs. Finally, Module #9 is high in C #3 with the expression of *PTP4A2, PTPN7,* and *CALM2* (Supplemental Fig. 7a).

After we defined the gene modules in each ILC3 cluster, we assessed the branch point in the trajectory analysis to determine genes that regulate the developmental program in ILC3 subsets in the LN (Fig. 7g & Supplemental Fig. 7b). The starting point of the pseudotime trajectory is followed by a major branch point (BP1), indicating two distinct cell fate decisions ($C_F$). Genes that are involved in BP and two $C_F$ can be defined as the modules. $C_{F1}$ is going toward C#2. Module #1, which contains highly expressed genes in $C_{F1}$, indicates transcripts for heat-shock proteins. In contrast, C #3 is mostly on the $C_{F2}$, and the genes involved in this module are mostly related to memory features, including *CCR7, SELL,* and *CD44*. Module #2 contains genes involved in the transition between the two cell fates and shows high expression before the branch point. These genes include *XCL2, LTB,* and *IL2*. The temporal expression of individual genes on the pseudotime axis also indicates that heat shock protein genes increase before C #2 is generated and remain high in this cluster. *CD48, CD52,* and *XCL2* genes that are expressed in C #1 are higher at the start point of pseudotime, and their expression decreases when they develop into C #2 or C #3. (Supplemental Fig. 7c). Collectively, our data demonstrate that the subsets of ILC1 and ILC3 are heterogeneous in human LNs and follow unique transcriptional and developmental programs.

## Discussion

In this study, we comprehensively examined the diverse subsets of ILCs present in human LNs. We found early development of ILCs can happen in the lymph nodes as a secondary lymphoid tissue. A naïve ILCs (nILCs) population exists in the human LNs that can differentiate into other ILC subtypes. We identified highly heterogeneous sub-clusters of ILC1 and ILC3 in both abdominal and thoracic human LNs. Compared to LNs, human spleens contained only minimal numbers of ILCs, albeit with similar heterogeneity. In contrast to LNs and spleen, human BM contained largely ILC precursors and lower numbers of mature ILCs.

The developmental process of ILCs in mice has been extensively reported, while limited information exists on the human[42,56]. ILC development in mice starts from HSCs in the BM[57] and then differentiates into lymphoid-primed multipotent progenitors (LMPPs) identified as Lin⁻Kit^Hi^Sca-1^Hi^Flt3⁺CD127⁻[58]. In humans, the development of ILCs starts

from BM with an ILC precursor population in circulation and tissue[59,60]. Interestingly, previous studies found LNs to be the site for the early development of NK cells. CD34^Dim^CD45RA⁺ α4B7⁺ subset resides in the parafollicular T cell regions and could develop into CD56^Bright^ NK cells when stimulated by IL-15 and IL-2 in vitro[61]. Our findings show that ILC precursors also exist in human LN, suggesting a role for the secondary lymphoid tissues in the early development of ILCs. The most immature ILC progenitors are also found in human tonsils[56,62]. These cells are termed early tonsillar progenitors (ETPs) and are defined as Lin⁻CD34⁺CD10⁺KIT⁻ (ETP1) and Lin⁻CD34⁺CD10⁻KIT⁺ (ETP2). ETP1 and IL-1R⁻ ETP2 are multipotential and could give rise to T cells and DCs under specific culture conditions. IL-1R1⁺ ETP2 lacks all non-NK/ILC potentials. Hence, IL-1R1⁺ ETP2 might be the earliest common ILC precursors to both NK cells and ILCs in humans[63]. These ETP cells in the human tonsil indicate the possibility of early developmental stages in the secondary lymphoid tissues. Our study identified the developmental stages of ILCs, from HSCs to ILC precursors in the BM. Although we found progenitor clusters in both LN and BM, the cells in the BM show higher expression of the genes related to hematopoiesis. The earliest common precursors for NK cells and ILCs are EILPs, divided into specified and committed EILPs[64]. While specified EILPs maintain residual DC potential, committed EILPs have lost DC potential and are progenitors dedicated to NK and ILC lineages (48). Our study defined two transitions from HSCs to CLPs and CLPs to EILPs in the BM. The next stage after EILP is the committed progenitors, including NKPs and pILCs. In mice, NKPs downstream of EILPs generate conventional NK cells, while CHILPs produce various types of ILCs, including LTi-like ILC3, ILC1, ILC2, and ILC3[65]. The GATA3⁺PLZF⁺ subset of CHILPs, known as ILCPs, has lost the ability to generate LTi-like ILC3 and produces ILC1, ILC2, and ILC3[5]. Consistent with our data, previous studies identified a pILC subset in mouse fetal liver and adult BM that transiently expressed a high amount of Plzf that were committed to with multiple ILC1, ILC2, and ILC3 potentials at the clonal level[4].

Previous studies uncovered a subset of naïve ILCs (nILCs) in the human colon that do not express master transcription factors while they express *IL7R* and *KIT*[82]. Trajectory analysis by Mazzurana et al. revealed potential differentiation pathways from naïve ILCs to three mature ILC3s (ILC3a-c) (28). ILC3a subset in the colon that expresses *XCL1* and *XCL2* is the first cluster in the trajectory. We discovered in our LN data that the first ILC3 cluster in the trajectory also expresses a high level of *XCL1* and *XCL2*. In addition to these two markers, our study indicates the expression of other genes in this cluster, including *KLF2, CTSW, EMP3, CLIC3,* and *LTB*. Our results show that the nILC population mainly exists in the LN but not in the BM or spl and expresses the genes upregulated on pILCs, including *ID2, ZBTB16,* and *IL-7R*. The comparison between gene regulatory networks of the nILC and the progenitor cluster indicates some unique activity of their regulons. Previous studies have shown that BCL11A is involved in early lymphoid development[49]. Consistent with that, BCL11A regulon is active in the BM progenitor population. IRF8 is another TF shown to be downregulated after being committed to mature ILCs. Consistent with that, IRF8 regulon is also active on the BM progenitor. On the contrary, the active regulons in the nILC population are related to effector functions such as NF-κB or ATF family. The presence of this naïve population could imply the necessity of fast response in these draining LN upon infection or disease condition. Our pseudotime trajectory analysis validates the link between the nILC population and all three ILCs.

We discovered extensive heterogeneity of ILCs in the human LNs. The transcriptomic and functional properties of ILC subsets in humans remain poorly understood. In humans, the ILC subtypes have been studied in lymphoid and non-lymphoid tissues, including blood, spleen, tonsil, adipose tissue, colon, and lungs[31,32,37,66]. Blood, spleen, and fat tissues have more ILC1 and ILC3 than ILC2[66]. Our findings demonstrate that ILC1, ILC2, and ILC3 exist within the human LN. Furthermore, we have observed that ILC1 and ILC3 are the most abundant subsets of ILCs in human LNs, whereas the proportion of ILC2 is relatively lower. This observation is similar to an earlier study by Mazzurana et al., who utilized full-length single-cell

sequencing on human colon samples and did not identify ILC2 as a major subset in the colon[37]. Recently, it has become evident that innate lymphoid cells (ILCs) are a highly diverse and complex population of innate immune cells. Based on the T-bet expression levels in ILC1 from mice intestines, they can be subdivided into four transcriptional states referred to as ILC1a-d subsets. The ILC1d subset expressed the highest T-bet levels[67]. Similarly, ILC3 in the mouse intestine is classified into five distinct subsets based on the expression of *NKp46, CCR6, CD49a*, and *CD4* transcripts[68]. In humans, ILC1 and ILC3 subsets were proportionally more frequent in mucosal and lymphoid sites, particularly among the colon and ileum IEL[66]. Recently, the Colonna group showed an additional population representing intermediate subsets of the ILC1 and ILC3[69]. They referred these intermediate populations as ILC3b (CD103⁺CD196⁺CD300LF⁺) and ILC1b (CD103⁺CD196⁺CD300LF⁻). In the current study, our transcriptomic profiling of the ILCs in LN identifies three subsets of ILC1, four subsets of ILC3, and one subset of ILC2. Bar- Ephraim et al. compared ILC3s in the LN, spl, and tonsil. Their data revealed that in the absence of inflammation, LN ILC3s do not express cytokine or their classical gene signature[25]. Along with their finding, our ILC3s from healthy individuals do not express NKP44 or the transcripts encoding cytokines IL-22 and IL-17A. In addition, several genes, including *IL18, IL1R1, TNFSF4*, and *IL7R*, are shared between our data and this earlier work[25].

ILC3 clusters include an early-stage expressing *XCL1, XCL2*, and *SELL*, the major subset in the BM. Our pseudotime trajectory of the BM ILCs validates this subset as the earliest cluster among developmental stages. There could be two possibilities for the presence of these clusters. These clusters could be the ILCs generated in BM and traffic to the LN. Another possibility is that these clusters are newly developed from nILC in the LN environment and not directly from BM. Our trajectory analysis supports the latter since the nILC mostly comes from the LN. This notion is further confirmed as pseudotime trajectory linked nILCs to ILC3 and ILC1. More in vitro analysis with sorted CXCR4⁺XCL1⁺XCL2⁺SELL⁺ ILC3 cells could unveil a better understanding of this phenomenon. ILC1 and ILC3 show a population with high expression of HSP transcripts, including *HSPA1B, HSPA1A, DNAJB1*, and *HSPE1*. Although HSPs form complexes with AHR, an important TF in ILC3 in normal conditions[70], these proteins are also expressed on the activated ILC2s[71]. HSP^high clusters in ILC1 and ILC3 could be the inflamed or active subset of ILCs in the LNs. In addition to HSP genes, these clusters express genes involved in inflammation, including *NEAT1, MALAT*, and *DEK*[72,73].

Our study comprehensively describes the transcriptional identity and developmental programs of the ILCs in human LNs. The insights gained in this study present a step toward a broader understanding of the tissue-specific development of ILCs and functions and create a roadmap for future studies to define new markers and potential pathways for therapeutic purposes.

## Methods

### Tissue collection and cell processing

LN and Spl samples were obtained from the Wisconsin Organ Donor Center, Versiti, WI. All samples from the organ donor center were provided anonymously after informed consent. The use of these human materials was approved by the IRB institutional IRB from Versiti. Versiti, Wisconsin, is one of 56 federally designated non-profit organ procurement organizations (OPO) in the country. Versiti Wisconsin's OPO facilitates deceased organ donation in 11 counties in eastern Wisconsin. The OPO's core responsibilities include donor family care and support, medical management and evaluation of organ donors, organ allocation and recovery, and public and professional education regarding the tremendous need for organ donation and transplantation. LN and Spl samples were received in the preserving media within 24 h after surgery, and we started the process immediately. Healthy human BM samples were de-identified samples. All fresh BM samples from healthy donors for the scRNA-seq and flow cytometry experiments were obtained from the Stem Cell and Xenograft Core of the University of Pennsylvania, PA. All samples were provided anonymously

after informed consent. The collection, distribution, and usage of these de-identified human materials were approved by the Institutional Review Board (IRB) of the University of Pennsylvania. The samples were shipped overnight and processed immediately upon receipt.

### Cell separation cell sorting

LN and Spleen tissues were kept in a complete RPMI medium, and fats were removed from the tissue. To make single-cell suspension, Multi-Tissue Dissociation kit from Miltenyi, biotech was used. 0.5 g of the tissue was cut into small pieces of 2–4 mm length and transferred into the GentleMACS C Tube containing the enzyme mix and RPMI-1640 medium on ice. GentleMACS Program for the tissue type of interest (i.e., 37 °C 'multi.B' preset program for human spleen) was selected. After completion, the aqueous layer was filtered into 50 ml conical tube using a 70 µm cellular strainer. Cells were centrifuged at 300 × g for 7 min and resuspended in 10 ml of complete RPMI media for counting. BM samples were collected in anticoagulated tubes and processed on the same day. Density gradient separation of BM samples was performed as follows. Samples were diluted with an equal volume of Dulbecco's phosphate buffered saline (PBS, Gibco) and layered over a density gradient medium (Lymphoprep, StemCell). Following centrifugation for 20 m at 800 × g, without brake, the mononuclear band was removed with a pipet, diluted, washed x 3 with PBS, and counted. Product samples were obtained directly from the manufactured Product. Washed with cold PBS + 0.4% bovine serum albumin (Sigma-Aldrich) and counted using a hemocytometer. Viability was assessed using trypan blue exclusion. Cells were allocated for FACS-sorting scRNAseq studies. All the innate lymphocytes were sorted as CD3E/CD19/CD20/CD14⁻CD7⁺ cells using FACSAria III or FACSMelody (BD Biosciences, San Jose, CA), and the purity was generally above 95%. Lin refers to CD3E, CD19, CD14, CD20. CD3E (UCHT1, 300417, @ 1:100), CD19 (HIB19, 302224, @ 1:100), CD14 (HCD14, 325616, @ 1:100), CD20 (2H7, 302320, @ 1:100), CD34 (581, 343512, @ 1:100), CD7 (CD7-6B7, 343108, @ 1:100), were from BioLegend (San Diego, CA).

### Single-cell RNA-seq for the sorted innate lymphocytes

**Library preparation and sequencing.** Cells were loaded on the 10X Chromium machine (10X Genomics) at target capture numbers of 8000–10000 viable cells/sample. Library construction was performed using the 10X Genomic Chromium Single Cell 3′ Reagent Kit v3.1 per manufacturer's protocols. Single-cell cDNA libraries were sequenced via Illumina Novaseq 6000 (Illumina) to a depth of around 50 million reads per cell. Raw data from each sample were demultiplexed after sequencing and aligned to NCBI Human Reference Genome Build GRCh38-3.0.0 reference genome, and UMI counts were quantified using 10X Genomics CellRanger software v3.0.0 using default parameters. Data were further processed using the Seurat package (v 3.2.2) in R (v3.5.1 or higher)[74].

**Quality Control.** QC steps included removing genes that expressed in three or fewer cells. Cells that expressed <100 or >6000 genes were filtered out to keep the good quality of cells. To remove apoptotic/lysing cells with high mitochondrial gene expression, we performed the filtering step to exclude cells with <2 or >20% mitochondrial gene expression. After removing the low-quality cells, a total of 163,057 cells were further analyzed. The average gene reads per cell was around 75,000 across all the samples.

**Clustering and cell-type Annotation.** All 14 samples from LNs, BM, and Spleens merged into a single Seurat object. The SCTransform function was used for normalization, finding highly variable features, and scaling. The variables UMI and mitochondrial transcript content were regressed to correct the batch effects. We used anchor-based integration to cluster the cells. Integration of different response groups at each time point was performed using the top 3000 features. Unbiased anchor-based clustering was then performed using the Seurat package's application of a shared nearest neighbor (SNN) modularity optimization-based

clustering algorithm (Louvain's original algorithm). The top 40 principal components were used for the analysis of the PCElbowPlot and Jack-StrawPlots. The number of principal components (PCs) for clustering analysis was picked as described before. The number reached the baseline of the standard deviation of PC. Dimensionality reduction was performed using principal component analysis (PCA) and uniform manifold and approximation projection (UMAP) embedding for visualization. Cell cluster resolution was set manually for each with the help of clustree analysis. Differential gene expression analysis (DEG) was performed using the non-parametric Wilcoxon rank sum test as implemented in FindMarkers function of the DESeq2 package v1.30.0. Cell identities were determined by using both known human NK cell-specific get sets and our database from published paper in our lab for NK clusters in BM and blood. Default parameters were used, including detection in a minimum of 10% of cells and at least a log fold change of 0.25. Correction for multiple tests was done using the Benjamini and Hochberg methods. As done above for the merged data from a single timepoint, cell cycle scoring was performed, and the SCTransform function was used for normalization, finding highly variable features and scaling; the variables UMI and mitochondrial transcript content were regressed out. Data were integrated using the top 3000 features, and anchor-based clustering was performed as described previously using PCA for dimensionality reduction and visualized using UMAP.

**Gene Set Enrichment Analysis**. (GSEA)[75,76] using the Reactome database[77] was performed by the gsePathway function in the ReactomePA package v1.34.0[21]. The gene list to conduct GESA was ranked by the logFC (log fold change) from DEGs in the FindMarkers function. Adjusted P values for GSEA were obtained after adjustment for multiple gene set enrichment comparisons. GSEA results were visualized using the ggplot2 package (version 3.3.5) and enrichplot package v1.10.1. Adjusted $P$ values < 0.05 were considered significant. Module score was calculated by using the AddModuleScore function with default parameters for different gene sets, which are from specific GSEA results and regulon results. Statistics of module scores are conducted by t-test, and $p < 0.05$ was considered significant.

**Monocle analysis**. Monocle 2 (version 2.14.0) and Monocle 3 (version 1.2.7) packages were used to view the cell state and trajectories in R[78,79]. The ordering genes for the cell clustering in the trajectory were picked up by the significant DEGs from different time points or different clusters in the Seurat object of FindAllMarkers with the cutoff of *p_val_adj < 0.05 & abs(aver_logFC) > 0.25*. The function of setOrderingFilter was used to order cells by the selected genes, and plot_ordering_genes was used to visualize the amount and location of the picked genes with the default parameter. The reduction_method is DDRTree. The function of plot_cell_trajectory was used to visualize the trajectory by time points, Seurat clusters, or Pesudotime.

**SCENIC analysis**. SCENIC package v1.1.2 was used to analyze the gene regulatory network[79,80]. The gene expression matrix was extracted, and cisTarget databases were 500 bp-upstream, and transcription start site-centered-10kb. (from https://resources.aertslab.org/cistarget/).

hg19_500bp_upstream_7species.mc9nr.feather and hg19_tss_centered_10kb_7species.mc9nr.feather. The binary regulon activity heatmap showed the regulons with absolute correlation >0.3 and active in at least 1% of cells. The co-expression target genes for a specific regulon were extracted to calculate the module score, visualized by feature plot. For the regulon activity on Seurat object, the matrix of 3.4_regulonAUC and the matrix from pData were combined. Plot the regulon activity on Seurat object by the function of plot_cell_trajectory.

**Reporting summary**
Further information on research design is available in the Nature Portfolio Reporting Summary linked to this article.

## Data availability

Data generated in this study are included in this published article and its supplementary information files. The data associated with this study are available from the corresponding author upon reasonable request. scRNA-seq data that support the findings of this study have been deposited in NCBI GEO with the #GSE243033 accession codes and are available at the following URL: https://www.ncbi.nlm.nih.gov/geo/query/acc.cgi?acc=GSE243033.

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

## Acknowledgements

This work was supported in part by NIH R01 AI102893 and NCI R01 CA179363 (SM); NCI R01 CA204231 (SR), HRHM Program of MACC Fund (SM and SR), Nicholas Family Foundation (SM); Gardetto Family (S.M.); and Every Day Good Foundation Inc (S.M.).

## Author contributions

E.H. and S.M. devised the project and performed the research. E.H. analyzed the data and wrote the manuscript. C.M. provide the human samples from The Wisconsin Organ Donor Center, Versiti. S.R. provided technical support for the experiments. S.M. and S.R. reviewed and edited the manuscript.

## Competing interests

The authors declare no competing interests.
