## [Peer Review File · Communications Biology]

Reviewers' comments:

Reviewer #1 (Remarks to the Author):

In the submitted manuscript, Hashemi et al. aimed to improve the current understanding of the diversity and developmental origins of ILCs in human lymph nodes, as compared to spleen and bone marrow. Over the last 10-20 years, the presence of ILC subsets in human secondary lymphoid tissues has been well described, and the authors sought to expand the knowledge of ILC heterogeneity particularly in human lymph nodes. The authors were able to successfully generate single cell RNA sequencing datasets from human lymph nodes, alongside spleen and bone marrow, from which they employed cluster analysis, pseudotime analysis, and DEG analysis. The authors identified three major groups of ILCs in lymph nodes and confirmed their presence via flow cytometry.

While a strength of this manuscript and a potentially important and novel contribution to the field is the human single cell RNA sequencing datasets generated, the reported results seem to be largely confirmatory of prior findings in the field and utilize previously defined definitions of various developmental stages of ILCs in addition to the three primary ILC subsets (ILC1, ILC2, ILC3). Although the authors were able to delineate heterogeneity among the ILC subsets based on pathway enrichment and DEGs, major novel roles or refinements of any ILC population identified seemed to be lacking. Furthermore, while the authors concluded that certain populations were immature and/or precursors to ILC subsets in the lymph nodes, they did not provide any functional/ex vivo confirmation through developmental assays or thorough identification with flow cytometry. In fact, their validation of the subsets identified with transcriptional data by flow cytometry was not sufficient, as the definitions were largely different and did not separate NK cells from ILC1s. Listed below are additional detailed concerns with the manuscript as presented in its current form:

1. The title of this manuscript focuses on addressing the diversity of ILCs within human lymph nodes. However, a key analysis seems to be missing, and that pertains to the characterization of possible differences in ILCs between the thoracic and abdominal lymph nodes. Currently, this manuscript largely focuses on lymph node ILCs as a collective compared to BM and spleen as determined by published gene sets and known pathway enrichment.
2. Some statements could likely be modified/softened so as to better acknowledge others' contributions in the field; for instance, lines 47-48 in the abstract. Indeed, the submitted study is certainly not the first analysis of human ILCs in lymph nodes; there are many such reports (e.g. Fehniger Blood 2003, Ferlazzo J Immunol 2004, Hoorweg Front Immunol 2012, and Bar-Ephraim Cell Rep 2017 to name a few).
3. Lines 85-86: the way this is worded implies that human ILC1s do not produce TNFalpha, yet they do (Kramer Cell Rep 2023).
4. Lines 98-99: the statement regarding Aiolos and T-bet in ILC3s/ILC1s in mice needs a reference.
5. Lines 116-117: thoracic lymph nodes would not be considered as mesenteric.

6. Regarding the initial evaluation of the CD7+ clusters, the authors stated that they removed clusters 6, 8, 9, and 10, because they represented B cells, red blood cells, and non-CD7-expressing cells, respectively. It stands to reason, then, that if such populations had contaminated the cellular preparations, there were very likely also contaminating T cells, which often outnumber B cells in tissues and – as opposed to B cells – normally express CD7. In light of this, it seems very skeptical that cluster #4 in the original cluster set (Figure 1) and clusters 0 and 1 in the refined clusters (after removing NK cells) consist of ILC1s and rather do not represent contaminating T cells. Indeed, they express many components of the CD3 receptor as well as TRAC, LEF1, TCF7, and/or BCL11B. None of the latter necessarily prove these cells are T cells, but the onus is on the authors to prove that they are not. What about expression of other T cell associated genes, such as CD5, CD26, CD28, TCR, and others? Can the authors definitively prove that these clusters in question do not represent T cells? And shouldn't ILC1s express TBX21/T-BET? The T-BET expression is apparently extremely low in these clusters and does not appear to be above what the other ILC clusters express. This is particularly important, because there are many in the field that are very skeptical of ILC1s existing as a distinct population in human lymphoid tissues, at least as defined by the Lin-CD127+CD117-CRTH2- phenotype used in the submitted manuscript. Others have demonstrated that such ILC1s do not exist as a distinct ILC1 population and rather frequently consist mostly if not entirely of T cells, ILC3s, ILC precursors, and NK cells (e.g. Simoni et al Immunity 2016, Chen Immunity 2018).

7. Also, regarding line 174 in the text: CD127, CD3E, and CD52 can each be expressed by NK cells; so, the argument made is not a sound one.

8. Line 203: KLRD1 is not NK-specific. Most liver ILC1s in mice (arguably the best well characterized ILC1 population described to date) express CD94/Nkg2a; and CD94 has also been described on most intraepithelial ILC1s (Fuchs Immunity 2013) and liver ILC1s in humans (Kramer Cell Rep 2023).

9. Regarding the discussion of HSCs, CLPs, etc in the bone marrow, what is the prior evidence that these populations are CD7+?

10. There are various typographical errors (e.g. line 406, line 250, etc), inconsistent use of subset names (e.g. immature ILCs vs nILCs, line 236-237), and inconsistencies in abbreviations (e.g. DEGs vs DE genes line 433/441). In addition, abbreviations should be clearly defined when first used in the text (including the abstract).

11. Fig. 1E shows quite a bit of sample to sample heterogeneity, particularly for the bone marrow samples. How did the authors validate that they had actually sampled bone marrow and not just blood?

12. It can be very helpful, especially to readers who may be color blind, to overlay cluster number designations on the corresponding clusters themselves in the UMAPs.

13. In Suppl Fig. 1B, it seems that the cutoffs for CD127+ cells are not consistent, as it looks like only for the bone marrow are CD127 weak+ cells included.

14. In Suppl Fig. 3B, only ILC1s are listed in the graph, but should the label not include NK cells as well

(based on what is shown in part A of that figure)?

15. Figure 4 highlights theoretical transitions of populations defined as CLPs to pILCS and then to ILC3s and ILC1s. In general, such analyses are useful for hypothesis generating but not as confirmatory approaches. Further, such findings reiterate already previously identified developmental stages of human ILCs and do not appear to be novel. In line 337, the authors speculate that new genes identified in cluster 4 of the LN and that are not present in the BM, suggestive of a developmental relationship between the pILCs in the LN and BM (another similar example is in line 353). Yet again no functional data are provided to support these theoretical developmental relationships.

16. If not already provided, it could be useful to include the sex and age for the donors from which each tissue sample was obtained.

Reviewer #2 (Remarks to the Author):

The manuscript by Hashemi et al. delineates an extensive and thorough analysis of mainly single cell sequencing data obtained from three thoracic-, three abdominal- lymph nodes (LN), four bone marrows (BM) and four spleens from healthy human donors. Based on this dataset with an impressive amount of cells, they have analysed the presence and trajectory of the ILCs within these tissues. They have also annotated specific regulatory genes (regulons) involved in deciding on a specific ILC lineage. The data presented on the human ILCs within these lymphoid tissues, notably not tonsil, is of high interest and provides an advance of our knowledge on the human ILCs.

Even though the manuscript is in essence very interesting, there are some points I would like to raise and which should be addressed;

1 Tissue collection:

- a) The tissues obtained were from different donors which were anonymised. However, age and sex of these donors should be known but was not presented, which is important to understand possible heterogeneity of the datasets. I can imagine that obtaining ILCs from LNs or BM from elderly donors will be different than from young donors. Also sex has been shown to be a major determinant in immune cell status. Subsequently, even though the authors show donor variation for the cell clusters (Fig. 1E), it is absolutely not clear how the authors assessed the differences in background of the donors vs. the differences in the cell clusters in Fig 1E. Especially BM1 seems to be an outlier, but I did not find information how this obvious differences in cell clusters between the donors were dealt with, or possible should have been excluded. Can these differences be attributed to a specific status of the donors? Please provide more information on the donors and the possible influence on the heterogeneity of the data.
- b) It is not clear whether the tissues obtained were from the same donor. I would strongly suggest to provide a table with the donors, background of the donor (at least age and sex), and the tissues collected from this donor in order to assess the heterogeneity due to tissue collection.
- c) what is the rationale to pool the thoracic and abdominal LNs? Was a comparison made between these LNs and if so, please state these data in the manuscript (possibly as supplemental).
- d) The introduction states that mucosal draining LNs have a special functioning in immune defence and tolerance induction (lines 107-112). Indeed, it has been shown before that differences between mucosal

vs peripheral LNs in human lead to different immune activation states of ILC3s (Bar-Ephraim et al., Cell Rep 2017, PMID: 29045847), especially of the tonsil vs other LNs. However, this publication and the notion of the mucosal draining LNs is not further assessed nor discussed. In my opinion, especially for the ILC3 analysis (Fig 7) it would be important to compare the data from this publication and discuss the immune status of the ILC(3).

2) The presentation of the data is sometime confusing and to me illogical. For example; the authors already state on line 233 that C#4 was mainly in the BM compared to LN, but they present the data in the next page on Fig4.

3) The cluster numbers are missing in the UMAP plots of Fig 2, 3, 4. The reference to Fig 2F on line 193 should be to fig 2E, to Fig2G on line 209 should be to 2F and fig 2G is not presented in the text. Fig 2E on line 189 is not present in the figure.

4) In the abstract it was stated that CD127+ ILCs were analysed (line 39), but most analysis have been done on CD7+ cells without a selection for CD127.

5) The authors frequently attribute a cell function to the expression of specific genes. Especially for CXCR4 this does not hold, as usually cells coming from the BM are CXCR4 high, but one of the main roles of CXCR4 is to adhere cells to the BM stroma. Regulation of CXCR4 function is very intricate and involves also regulation of the receptor on the surface. Please tune down statements or suggestions on gene expression with cell functioning (eg lines 419, 427, 439, 459).

6) line 90- PLZF is the protein, thus not italics. line 91 ZBTB16 is the gene, thus italics

7) ILC3 cluster in Fig 3 is situated between the CLP and nILC and in Fig 4 between the nILC and the other ILCs. Also, the pseudo time analysis indicate a progression through this cluster. Can the authors elaborate on this phenomenon, what does it tell about the ILC3? Would this analysis suggest that the other ILCs could be derived from the ILC3 population? Or can this be attributed to the algorithm used and genes selected?

8) The progenitor ILC (pILC) has been discussed in the text of Fig 3 (lines 278. 288 (with EILP), 288), but this population has not been annotated in the clusters in the UMAP plots.

9) I did not understand the analysis and heat map presented in fig 5 E and associated conclusions on regulons.

Point-to-point Reply

COMMSBIO-23-4093-T: Transcriptomic Diversity of Innate Lymphoid Cells in Human Lymph nodes

We thank the Reviewers and the Editor for the constructive comments. Please find our answers to the Reviewer's comments below. We have addressed all the concerns from both the Reviewers. The changes and additions are listed here and in the manuscript text. We have extensively modified the text of our manuscript following the Reviewers' suggestions. Please see the highlighted text in the manuscript.

Reviewer #1 (Remarks to the Author):

In the submitted manuscript, Hashemi et al. aimed to improve the current understanding of the diversity and developmental origins of ILCs in human lymph nodes, as compared to spleen and bone marrow. Over the last 10-20 years, the presence of ILC subsets in human secondary lymphoid tissues has been well described, and the authors sought to expand the knowledge of ILC heterogeneity particularly in human lymph nodes. The authors were able to successfully generate single cell RNA sequencing datasets from human lymph nodes, alongside spleen and bone marrow, from which they employed cluster analysis, pseudotime analysis, and DEG analysis. The authors identified three major groups of ILCs in lymph nodes and confirmed their presence via flow cytometry.

While a strength of this manuscript and a potentially important and novel contribution to the field is the human single-cell RNA sequencing datasets generated, the reported results seem to be largely confirmatory of prior findings in the field and utilize previously defined definitions of various developmental stages of ILCs in addition to the three primary ILC subsets (ILC1, ILC2, ILC3). Although the authors were able to delineate heterogeneity among the ILC subsets based on pathway enrichment and DEGs, major novel roles or refinements of any ILC population identified seemed to be lacking. Furthermore, while the authors concluded that certain populations were immature and/or precursors to ILC subsets in the lymph nodes, they did not provide any functional/ex vivo confirmation through developmental assays or thorough identification with flow cytometry. In fact, their validation of the subsets identified with transcriptional data by flow cytometry was not sufficient, as the definitions were largely different and did not separate NK cells from ILC1s. Listed below are additional detailed concerns with the manuscript as presented in its current form:

1. The title of this manuscript focuses on addressing the diversity of ILCs within human lymph nodes. However, a key analysis seems to be missing, and that pertains to the characterization of possible differences in ILCs between the thoracic and abdominal lymph nodes. Currently, this manuscript largely focuses on lymph node ILCs as a collective compared to BM and spleen as determined by published gene sets and known pathway enrichment.

We thank the Reviewer for this suggestion. We changed the title to "Transcriptomic Diversity of Innate Lymphoid Cells in Human Lymph Nodes compared to BM and Spleen" to avoid misinterpretation.

We have included two heatmaps in the **Supplementary Figure 6** and added explanation in the text of **updated manuscript**. In the first heatmap, we emphasize the similarity of LN ILCs compared to BM and Spleen. The second heatmap shows the top differentially expressed genes between thoracic and abdominal lymph nodes.

2. Some statements could likely be modified/softened so as to better acknowledge others' contributions in the field; for instance, lines 47-48 in the abstract. Indeed, the submitted study is certainly not the first analysis of human ILCs in lymph nodes; there are many such reports (e.g. Fehniger Blood 2003, Ferlazzo J Immunol 2004, Hoorweg Front Immunol 2012, and Bar-Ephraim Cell Rep 2017 to name a few).

We agree with the Reviewer. We have changed the sentences in the current version of the manuscript.

Our new findings are:

- 1- Although the subgroup of ILCs in the tonsil or colon has been defined, this characterization was not performed for the LNs. In the current study, we define three subgroups of ILC1 and four subgroups of ILC3 with unique transcriptomic profiles.
- 2- We compared the LN ILC to BM as primary and spleen as secondary lymphoid organs. We also link the BM progenitors to naïve ILCs in the LNs.

3. Lines 85-86: the way this is worded implies that human ILC1s do not produce TNF-alpha, yet they do (Kramer Cell Rep 2023).

We agree with the Reviewer. Based on Kramer *et al.*, liver-type ILC1 produces TNF- α . Therefore, we modified the sentences.

The modified sentences are: "ILC1 produces interferon-gamma (IFN- γ), and they require T-BET for their differentiation⁹. While NK cells also need T-BET to develop and function, they are highly cytolytic by producing granzymes (GZMs), including GZM-B and GZM-H."

4. Lines 98-99: the statement regarding Aiolos and T-bet in ILC3s/ILC1s in mice needs a reference. Thank you. We have added the Luca Mazzurana *et al. Eur J Immunology* 2019 in the updated text.

5. Lines 116-117: thoracic lymph nodes would not be considered as mesenteric. We apologize for the mistake. We corrected this in the current version of the manuscript.

6. Regarding the initial evaluation of the CD7+ clusters, the authors stated that they removed clusters 6, 8, 9, and 10, because they represented B cells, red blood cells, and non-CD7-expressing cells, respectively. It stands to reason, then, that if such populations had contaminated the cellular preparations, there were very likely also contaminating T cells, which often outnumber B cells in tissues and? as opposed to B cells? normally express CD7. In light of this, it seems very skeptical that cluster #4 in the original cluster set (Figure 1) and clusters 0 and 1 in the refined clusters (after removing NK cells) consist of ILC1s and rather do not represent contaminating T cells. Indeed, they express many components of the CD3 receptor as well as TRAC, LEF1, TCF7, and/or BCL11B. None of the latter necessarily prove these cells are T cells, but the onus is on the authors to prove that they are not. What about expression of other T cell associated genes, such as CD5, CD26, CD28, TCR, and others? Can the authors definitively prove that these clusters in question do not represent T cells?

- 1- We appreciate the Reviewer's concern. We used two methods to make sure we were analyzing only ILCs. First, we used an anti-CD3 ϵ antibody and other B and myeloid lineage-specific antibodies in our sorting to exclude non-ILCs. Second, we excluded #6, #8, #9, and #10 from our first set of analyses because these clusters expressed transcripts representing red blood cells and B cells, which were transcriptomically distinct from the main ILC clusters. Their transcriptomic profiles indicated they did not belong to ILCs, and, therefore, they were removed.
- 2- In the remaining clusters, we have validated that they are not T-cells by checking the expression of CD4 and CD8 and the other markers the Reviewer asked to check. None of these T-cell-associated genes are differentially expressed on any of these clusters. Below is the list of gene expressions presented as violin plots. TRB, TRD, and TRG transcripts could not be detected in our scRNA-seq data set.
- 3- Earlier studies have shown that the TRAC and CD3 transcripts are present in the ILC1 subset; these genes alone cannot be used as markers to distinguish between T and ILCs. However, several other markers, including CD28, CD5, CD4, CD8B, CD8A, RAG1, RAG2, TRGC2, and DPP4

were not expressed in clusters #4 and #5, which we identify as ILCs. Please see the above figure for the expression of these genes.

2 - To address the possible contamination of T cells in some ILC clusters.

- 1- We appreciate the Reviewer's concern. We used two methods to make sure we were analyzing only ILCs. **First**, we used an anti-CD3 ϵ antibody and other B and myeloid lineage-specific antibodies in our sorting to exclude non-ILCs. **Second**, we excluded #6, #8, #9, and #10 from our first set of analyses because these clusters expressed transcripts representing red blood cells and B cells, which were transcriptomically distinct from the main ILC clusters. Their transcriptomic profiles indicated they did not belong to ILCs, and, therefore, they were removed.
- 2- In the remaining clusters, we have validated that they are not T-cells by checking the expression of CD4 and CD8 and the other markers the Reviewer asked to check. None of these T-cell-associated genes are differentially expressed on any of these clusters. Below is the list of gene expressions presented as violin plots. TRB, TRD, and TRG transcripts could not be detected in our scRNA-seq data set. Therefore, we conclude no T cell contamination in our clusters.

Figure A. For review purpose only. **Minimal or no expression levels of T cell-associated gene transcripts in ILC clusters.** To exclude the possibility of T cell contamination in ILC clusters, we analyzed several transcripts. Top three rows represent T cell-associated genes. We did not find any transcript levels that convincingly support the presence of T cells in the eight clusters. The bottom row represents gene transcripts that are shared or expected to be present in ILCs. We included them here as positive controls.

Earlier studies have shown that the TRAC and CD3 transcripts are present in the ILC1 subset; these genes alone cannot be used as markers to distinguish between T and ILCs. However, several other markers, including CD28, CD5, CD4, CD8B, CD8A, RAG1, RAG2, TRGC2, and DPP4 were not expressed in clusters #4 and #5, which we identify as ILCs. Please see the above figure for the expression of these genes.

And shouldn't ILC1s express TBX21/T-BET? The T-BET expression is apparently extremely low in these clusters and does not appear to be above what the other ILC clusters express. This is particularly important, because there are many in the field that are very skeptical of ILC1s existing as a distinct population in human lymphoid tissues, at least as defined by the Lin-CD127+CD117-CRTH2- phenotype used in the submitted manuscript. Others have demonstrated that such ILC1s do not exist as a distinct ILC1 population and rather frequently consist mostly if not entirely of T cells, ILC3s, ILC precursors, and NK cells (e.g. Simoni et al Immunity 2016, Chen Immunity 2018).

We thank the Reviewer for the comment.

- 1- We agree that ILC1 expresses TBX21. We do find *TBX21* transcripts in our data. The following dot plot shows the expression level in the combined data from NK and ILCs. *TBX21* is highly expressed in NK cells. *TBX21* is also expressed at lower levels in Clusters #4 and #5, which contain ILC1.

- 2- Simoni *et al.*, 2016 identified NK, ILC2, ILC3, and intra-epithelial ILC1-like cells in different tissues using CyTOF. However, other studies employing scRNA-seq showed the existence of the ILC1 as a major subgroup among the total ILCs, including Mjösberg *et al.* *Nat Immunol* 2016. They defined all three subsets of ILCs and validated the existence of ILC1 using a unique gene set. Thus, we agree considering ILC1 as a distinct subset is controversial in the field.

Irrespective of these observations, while none of these studies used human lymph nodes to define subsets of ILCs, we identify a subset of cells with ILC1 characteristics, such as CD3 and CD127 transcripts, with minimal and no expression of T cell-defining transcripts (as indicated by the above violin plots).

7. Also, regarding line 174 in the text: CD127, CD3E, and CD52 can each be expressed by NK cells; so, the argument made is not a sound one.

We agree with the Reviewers that these markers could be expressed in NK cells. Although these markers can be expressed in NK cells, the relative levels of these transcripts are lower in mature NK cells compared to ILCs, and earlier studies used these markers to validate ILC vs NK cells (Åsa K Björklund *et al.*, and Luca Mazzurana *et al.*).

<https://www.nature.com/articles/s41422-020-00445-x>

<https://www.nature.com/articles/ni.3368>

8. Line 203: KLRD1 is not NK-specific. Most liver ILC1s in mice (arguably the best well characterized ILC1 population described to date) express CD94/Nkg2a; and CD94 has also been described on most intraepithelial ILC1s (Fuchs Immunity 2013) and liver ILC1s in humans (Kramer Cell Rep 2023).

We agree with the Reviewer that these genes are not NK-specific, and our statement that only NK cells express these genes could be misleading. Therefore, we change that to “NK-associated genes” instead of “NK-specific” genes.

9. Regarding the discussion of HSCs, CLPs, etc in the bone marrow, what is the prior evidence that these populations are CD7+?

The following studies demonstrate the expression of CD7 on CLPs:

In the study by Q L Hao et al. 2001, a clonogenic subpopulation of CD34⁺CD38⁻ in cord blood cells that expressed high levels of CD7 possessed only lymphoid potential was identified. These cells uniformly expressed CD45RA and could generate NK cells and other lymphocytes.

In the second study, Anne H.M. Galy et al. 1995, identified the presence of CD7⁺ T cell progenitors among CD34⁺ fetal bone marrow and fetal liver cells.

The third study by Priyanka Sathe indicates that lineage potential subsets within the CD7⁺ fraction are based on CD127 and CD117 expression, which possessed restricted lymphoid and biased NK cell lineage potentials.

<https://pubmed.ncbi.nlm.nih.gov/11389003/>

<https://www.sciencedirect.com/science/article/pii/S0006497120764499?via%3Dihub>

<https://www.ncbi.nlm.nih.gov/pmc/articles/PMC5751317/>

10. There are various typographical errors (e.g. line 406, line 250, etc), inconsistent use of subset names (e.g. immature ILCs vs nILCs, line 236-237), and inconsistencies in abbreviations (e.g. DEGs vs DE genes line 433/441). In addition, abbreviations should be clearly defined when first used in the text (including the abstract).

We thank the Reviewer for the comment. We have corrected these typographical errors in the current version of the manuscript.

11. Fig. 1E shows quite a bit of sample-to-sample heterogeneity, particularly for the bone marrow samples. How did the authors validate that they had actually sampled bone marrow and not just blood? The sampling is based on the clinical instruction from the bones.

We used fresh bone marrow samples from four healthy donors obtained by the Stem Cell and Xenograft Core of the University of Pennsylvania, PA. After we isolated mononuclear cells using lymphoprep, we sorted the CD7⁺ cells and performed scRNA sequencing for individual samples from each donor. Sequencing and quality control were done separately on each sample, and then the data was combined to unsupervised clustering using R-packages. We did not use any peripheral blood mononuclear samples (PBMCs) in this study.

Although the samples have some level of variation, all the BM samples showed similarity compared to LN or Spleen samples. Cluster #0 is the mature NK cells consistent between the BM samples and is the highest cluster. Cluster #1 also is mature NK cells with a higher percentage in one of the individuals (Cluster #4 and #5). We did not remove this individual as the variation is mostly on one subset of the NK cells.

12. It can be very helpful, especially to readers who may be color blind, to overlay cluster number designations on the corresponding clusters themselves in the UMAPs.

We thank the Reviewer for the suggestion. We have included cluster numbers in all the UMAPs.

13. In Suppl Fig. 1B, it seems that the cutoffs for CD127⁺ cells are not consistent, as it looks like only for the bone marrow are CD127 weak⁺ cells included.

We thank the reviewer for the valid comment. We have changed the gating in these flow plots for all the samples and calculated the percentage based on the changes. The updated figure is shown in **Supplementary Figure 1B and 1C**.

14. In Suppl Fig. 3B, only ILC1s are listed in the graph, but should the label not include NK cells as well (based on what is shown in part A of that figure)?

Yes. We corrected that in the updated version of the manuscript **Figure. 3B**.

15. Figure 4 highlights theoretical transitions of populations defined as CLPs to pILCS and then to ILC3s and ILC1s. In general, such analyses are useful for hypothesis generating but not as confirmatory approaches. Further, such findings reiterate already previously identified developmental stages of human ILCs and do not appear to be novel. In line 337, the authors speculate that new genes identified in cluster 4 of the LN and that are not present in the BM, suggestive of a developmental relationship between the pILCs in the LN and BM (another similar example is in line 353). Yet again no functional data are provided to support these theoretical developmental relationships.

We agreed with the reviewer that these populations have been identified before. Here, we show the transitions of populations with gene expressions associated with known progenitor populations. Although the steps of progenitor populations have been already defined, the progression from BM to LNs and its connection to naïve ILCs has not been defined. We also modified **Figure 4** in the updated manuscript.

The second comment of the reviewer is valid in that we did not provide functional data, as our analyses primarily focused on developing ILCs. We also express our inability to perform functional analyses on these samples as the number of cells from each donor is limited.

16. If not already provided, it could be useful to include the sex and age for the donors from which each tissue sample was obtained.

We thank the Reviewers for this suggestion. We now included a Supplementary Table containing the age and sex of all donors. For each tissue, we obtained equal numbers of females and males. We had two females and two males for the BM. Samples from three females and three males were used for the LN ILC analyses. Samples from two females and two males were used for the spleen ILC analyses. The ages of the donors ranged between 19- 60 years.

Reviewer #2 (Remarks to the Author):

The manuscript by Hashemi et al. delineates an extensive and thorough analysis of mainly single cell sequencing data obtained from three thoracic-, three abdominal- lymph nodes (LN), four bone marrows (BM) and four spleens from healthy human donors. Based on this dataset with an impressive amount of cells, they have analysed the presence and trajectory of the ILCs within these tissues. They have also annotated specific regulatory genes (regulons) involved in deciding on a specific ILC lineage. The data presented on the human ILCs within these lymphoid tissues, notably not tonsil, is of high interest and provides an advance of our knowledge on the human ILCs.

Even though the manuscript is in essence very interesting, there are some points I would like to raise and which should be addressed;

1 Tissue collection:

a) The tissues obtained were from different donors which were anonymised. However, age and sex of these donors should be known but was not presented, which is important to understand possible heterogeneity of the datasets. I can imagine that obtaining ILCs from LNs or BM from elderly donors will be different than from young donors. Also, sex has been shown to be a major determinant in immune cell status. Subsequently, even though the authors show donor variation for the cell clusters (Fig. 1E), it is absolutely not clear how the authors assessed the differences in background of the donors vs. the differences in the cell clusters in Fig 1E. Especially BM1 seems to be an outlier, but I did not find information how this obvious differences in cell clusters between the donors were dealt with, or possible should have been excluded. Can these differences be attributed to a specific status of the donors? Please provide more information on the donors and the possible influence on the heterogeneity of the data.

We thank the Reviewers for this suggestion. We now included a Table containing information about the age and sex of all donors. For each tissue, we obtained equal numbers of females and males. We used BM from two females and two males. LNs were obtained from three females and three males. Spleens from two females and two males were used. The ages of the donors ranged between 19- 60 years. The following table is added as supplementary material to the manuscript.

Sample	Age	Gender
BM1	19	Female
BM2	25	Female
BM3	28	Male
BM4	44	Male
LN-A1	15	Female
LN-A2	53	Female
LN-A3	51	Male
LN-T1	56	Female
LN-T2	51	Male
LN-T3	60	Male
Spl1	51	Male
Spl2	56	Male
Spl3	54	Female
Spl4	53	Female

Although the samples have some level of variation, all the BM samples showed similarity compared to LN or Spleen samples. Cluster #0 is the mature NK cells that are consistent with the BM samples and contain the highest number of NK cells. Cluster #1 also has mature NK cells with a higher percentage in one of the individuals. We did not remove this donor because we focused on the ILC subsets in this study (Cluster #4 and #5) as the variation is mostly on one subset of the NK cells.

b) It is not clear whether the tissues obtained were from the same donor. I would strongly suggest to provide a table with the donors, background of the donor (at least age and sex), and the tissues collected from this donor in order to assess the heterogeneity due to tissue collection.

Spleens and LNs were obtained from the Versiti Wisconsin Organ Donor Center, Milwaukee, WI. Two donors provided both the spleens and the LNs. The remaining were from independent donors. Bone marrows were obtained from unrelated living donors from the Stem Cell and Xenograft Core of the University of Pennsylvania, PA. Please see the color coding in **Supplementary Table 1** that indicates the related donors.

c) what is the rationale to pool the thoracic and abdominal LNs? Was a comparison made between these LNs and if so, please state these data in the manuscript (possibly as supplemental).

We did not pool the samples. We received three abdominal and three thoracic samples and processed them independently. After we dissociated the cells using the Multi-Tissue Dissociation kit from Miltenyi Inc., we sorted the CD7⁺ cells and performed scRNA sequencing for each sample. Sequencing and quality control were done separately on each sample, and then the data was combined for unsupervised clustering using R-packages.

As per the Reviewer's suggestion, we now compared the thoracic and abdominal lymph nodes. Please see the additional data presented in **Supplementary Figure 6**. First, we compared the ILCs from LN samples to BM and Spleen. Second, we compared the top DE genes that differ between thoracic vs abdominal ILCs.

d) The introduction states that mucosal-draining LNs have a special functioning in immune defense and tolerance induction (lines 107-112). Indeed, it has been shown before that differences between mucosal vs peripheral LNs in human lead to different immune activation states of ILC3s (Bar-Ephraim et al., Cell Rep 2017, PMID: 29045847), especially of the tonsil vs other LNs. However, this publication and the notion of the mucosal draining LNs is not further assessed nor discussed. In my opinion, especially for the ILC3 analysis (Fig 7) it would be important to compare the data from this publication and discuss the immune status of the ILC(3).

We thank the Reviewer for the suggestion. Data from Bar-Ephraim *et al.* revealed that in the absence of inflammation, LN ILC3s do not express cytokine or gene signatures normally associated with ILC3s.

Along with their finding, our ILC3s from LNs of healthy individuals are mostly in the resting state. Consistent with this, LN ILC3s from our data also do not express NKP44 or transcripts encoding IL-22 and IL-17A.

In addition to that, as shown in their Figure 3 heatmap, they compared LN ILC3 to the spleen or tonsil. Both their and our data identified a common gene signature that included IL-18, IL1R1, TNFSF4, and IL7R. We have now summarized and included this information in the current version of our manuscript (**Discussion** section). We have also included this reference in our paper.

2) The presentation of the data is sometime confusing and to me illogical. For example; the authors already state on line 233 that C#4 was mainly in the BM compared to LN, but they present the data in

the next page on Fig 4.

We apologize for the confusion. We have modified the text to avoid any confusion. **Figure 3** is the first place we mentioned Cluster #4 is higher in the BM. But the statement is based on the UMAP in the combined data, while in **Figure 4C** we separated Cluster #4 and validated that using dotplot.

3) The cluster numbers are missing in the UMAP plots of Fig 2, 3, 4. The reference to Fig 2F on line 193 should be to fig 2E, to Fig2G on line 209 should be to 2F and fig 2G is not presented in the text. Fig 2E on line 189 is not present in the figure.

We appreciate the useful comment by the Reviewer. We have added cluster numbers in the updated figures. The figure number was corrected in the text, as mentioned.

4) In the abstract it was stated that CD127+ ILCs were analyzed (line 39), but most analysis have been done on CD7+ cells without a selection for CD127.

We agree with the Reviewer. We sorted the CD7⁺ cells, but after performing single-cell RNA sequencing, we isolated CD127⁺ cells as non-NK cells (Clusters #4 and # 5) and identified different subsets in these populations. To avoid discrepancies in the abstract, we changed the sentence to “Here, we isolated CD7⁺ cells and performed single-cell RNA sequencing to unravel the tissue-specific transcriptional profiles of 47,287 CD127⁺ ILCs within the human abdominal and thoracic LNs.”

5) The authors frequently attribute a cell function to the expression of specific genes. Especially for CXCR4 this does not hold, as usually cells coming from the BM are CXCR4 high, but one of the main roles of CXCR4 is to adhere cells to the BM stroma. Regulation of CXCR4 function is very intricate and involves also regulation of the receptor on the surface. Please tune down statements or suggestions on gene expression with cell functioning (eg lines 419, 427, 439, 459).

We thank the Reviewer for the comment. We have modified these sentences in the current version of the manuscript.

6) line 90- PLZF is the protein, thus not italics. line 91 ZBTB16 is the gene, thus italics
We apologize for the error. We have corrected these in the updated version of the manuscript.

7) ILC3 cluster in Fig 3 is situated between the CLP and nILC and in Fig 4 between the nILC and the other ILCs. Also, the pseudotime analysis indicates a progression through this cluster. Can the authors elaborate on this phenomenon, what does it tell about the ILC3? Would this analysis suggest that the other ILCs could be derived from the ILC3 population? Or can this be attributed to the algorithm used and genes selected?

Although there is evidence about the plasticity between ILC3-ILC1 (Bernink JH et al. *Immunity* 2015 and Carlo De Salvowe, *Frontiers in Immunology* 2020), we still think the result in Pseudotime trajectory is mostly due to the similarity of the gene expression.

We could not exclude the possibility of the differentiation of ILC1 and ILC2 from ILC3. We also think this Pseudotime analysis algorithm is needed to define a link between the cells, and based on more overlapped genes between nILC and ILC3, the trajectory passed the ILC3 first. The site of cluster ILC3 being closer to nILC in the UMAP plot also confirms this.

8) The progenitor ILC (pILC) has been discussed in the text of Fig 3 (lines 278, 288 (with EILP), 288), but this population has not been annotated in the clusters in the UMAP plots.

Since the progenitor cluster was small, the subpopulation was not separated as an individual cluster. Therefore, we changed the writing in the text to this cluster as the progenitor population. Both pILC and EILP are in the progenitor cluster defined as Cluster #4, and their locations will be discussed in the 4E and 4F.

9) I did not understand the analysis and heat map presented in fig 5 E and associated conclusions on regulons.

This heatmap shows the different regulons active in nILCs in the LN and progenitor population in the BM. This panel emphasizes how the regulons function through the ILC development. One of the examples is regulons related to Immediate early genes (IEGs) and NF- κ B pathway that are active in the nILCs.

Reviewers' comments:

Reviewer #1 (Remarks to the Author):

The revised manuscript is substantially improved in response to my (and the other reviewer's) original concerns. I think that the authors have done very well to add important clarity and balance where needed and to "soften" some of the original claims; and they were also able to provide an insightful, thorough, and novel analysis of the complex nature of ILCs in various lymphoid tissues. This will likely serve as a great resource for the field. I have no further concerns and congratulate the authors on their work.

Reviewer #2 (Remarks to the Author):

I would like to express my gratitude to the authors for addressing the queries and enhancing the manuscript. However, I would like to bring to your attention some concerns that I believe still require careful consideration.

Firstly, while I acknowledge the improvements made, I remain unsatisfied with certain responses. The other reviewer has also pointed out discrepancies, some of which were not fully rectified in the revised version. For instance, variations like "DEGs" and "DE genes" persist, and the cluster numbers remain absent in several UMAP graphs (specifically in 2A, 3A, 4D), despite the authors' assurance of addressing these matters in the rebuttal. Additionally, on line 305, the reference to the SOX4 population should correspond to Fig 4E. I kindly suggest a thorough review of the manuscript to address these minor yet persistent inaccuracies.

In addition to the unresolved issues, the authors indicated an intention to incorporate relevant literature on human ILC analysis in the introduction (paragraph 4). While the publication of Bar-Ephraim is discussed in the discussion section, it is not referenced, and there is a need to include it and any other relevant publications mentioned by the other reviewer in the introduction. This would provide context and allow the authors to articulate what was lacking in earlier analyses and how the current study addresses those gaps (top of page 2 of your rebuttal letter).

Specific points;

Point 3; Although they have now referred to the figures correctly, the referred order of the figures in the text does not match the order of appearance in the figures.

Point 7: I still find the pseudotime analysis challenging to understand. The description in the text of the transition from CLP to pILC to ILC3 and ILC1, does not completely align with the trajectory shown in Fig. 4D. Clarification on the authors' logic and additional details in the discussion (lines 601-608) would greatly aid in understanding. Elaborating on the trajectory and how it supports the suggestion that "these clusters are newly developed from nILC in the LN environment and not directly from BM" (line 604-605) would be beneficial. Additionally, considering excluding BM cells from the trajectory analysis for LN/spleen cells might enhance the clarity of this point.

Point-to-point Reply
COMMSBIO-23-4093-T

Transcriptomic Diversity of Innate Lymphoid Cells in Human Lymph Nodes compared to BM and Spleen

Reviewer #1 (Remarks to the Author):

The revised manuscript is substantially improved in response to my (and the other reviewer's) original concerns. I think that the authors have done very well to add important clarity and balance where needed and to "soften" some of the original claims; and they were also able to provide an insightful, thorough, and novel analysis of the complex nature of ILCs in various lymphoid tissues. This will likely serve as a great resource for the field. I have no further concerns and congratulate the authors on their work.

We thank the Reviewer.

Reviewer #2 (Remarks to the Author):

I would like to express my gratitude to the authors for addressing the queries and enhancing the manuscript. However, I would like to bring to your attention some concerns that I believe still require careful consideration.

1. Firstly, while I acknowledge the improvements made, I remain unsatisfied with certain responses. The other reviewer has also pointed out discrepancies, some of which were not fully rectified in the revised version. For instance, variations like "DEGs" and "DE genes" persist,

We apologize for the mistake. We made sure that we corrected this in the current version of the manuscript.

2. The cluster numbers remain absent in several UMAP graphs (specifically in 2A, 3A, 4D), despite the authors' assurance of addressing these matters in the rebuttal.

We added the cluster numbers in Figures 2A, 3A, 4D in this revised version of the manuscript.

3. Additionally, on line 305, the reference to the SOX4 population should correspond to Fig 4E. I kindly suggest a thorough review of the manuscript to address these minor yet persistent inaccuracies.

Thank you. We corrected this in the current version of the manuscript (Line# 303).

4. In addition to the unresolved issues, the authors indicated an intention to incorporate relevant literature on human ILC analysis in the introduction (paragraph 4). While the

publication of Bar-Ephraim is discussed in the discussion section, it is not referenced, and there is a need to include it and any other relevant publications mentioned by the other reviewer in the introduction. This would provide context and allow the authors to articulate what was lacking in earlier analyses and how the current study addresses those gaps (top of page 2 of your rebuttal letter).

Thank you for your suggestion. We have added these references (Line# 580-584, in blue) and other related references in the introduction (Line 109-112, references).

5. Specific points;

Point 3; Although they have now referred to the figures correctly, the referred order of the figures in the text does not match the order of appearance in the figures.

We have corrected the orders of the Figure numbers matching with the updated text in the current version of the manuscript.

Point 7: I still find the pseudotime analysis challenging to understand. The description in the text of the transition from CLP to pILC to ILC3 and ILC1, does not completely align with the trajectory shown in Fig. 4D. Clarification on the authors' logic and additional details in the discussion (lines 601-608) would greatly aid in understanding. Elaborating on the trajectory and how it supports the suggestion that "these clusters are newly developed from nILC in the LN environment and not directly from BM" (line 604-605) would be beneficial. Additionally, considering excluding BM cells from the trajectory analysis for LN/spleen cells might enhance the clarity of this point.

Thank you for suggestion. The Figure 4 is only the BM showing a link between the pILC and mature ILCs. BM did not contain nILCs; therefore, we excluded only LNs (as you suggest in this comment) and showed the direct link from nILC to ILC3 and ILC1 in the LNs (supplement Figure-5). We modified the text in the discussion (Line# 586-591) in updated version.

REVIEWERS' COMMENTS:

Reviewer #2 (Remarks to the Author):

I thank the authors for their efforts to correct the manuscript. I have no further comments.